# Cloud Photogrammetry with Dense Stereo for Fisheye Cameras

Christoph Beekmans[1], Johannes Schneider[2], Thomas Läbe[2], Martin Lennefer[1], Cyrill Stachniss[2], and Clemens Simmer[1]

[1]Meteorological Institute, University of Bonn
[2]Institute of Geodesy and Geoinformation, University of Bonn

*Correspondence to:* Christoph Beekmans (cbeekmans@uni-bonn.de)

**Abstract.** We present a novel approach for dense 3D cloud reconstruction above an area of $10 \times 10$ km$^2$ using two hemispheric sky imagers with fisheye lenses in a stereo setup. We examine an epipolar rectification model designed for fisheye cameras, which allows the use of efficient out-of-the-box dense matching algorithms designed for classical pinhole-type cameras to search for correspondence information at every pixel. The resulting dense point cloud allows to recover a detailed and more complete cloud morphology, compared to previous approaches that employed sparse feature-based stereo or assumed geometric constraints on the cloud field. Our approach is very efficient and can be fully automated. From the obtained 3D shapes, cloud dynamics, size, motion, type and spacing can be derived and used e.g. for radiation closure under cloudy conditions.

Fisheye lenses follow a different projection function than classical pinhole-type cameras and provide a large field of view with a single image. However, the computation of dense 3D information is more complicated and standard implementations for dense 3D stereo reconstruction cannot be easily applied.

Together with an appropriate camera calibration, which includes internal camera geometry and global position and orientation of the stereo camera pair, we use the correspondence information from the stereo matching for dense 3D stereo reconstruction of clouds located around the cameras.

We implement and evaluate the proposed approach using real world data and present two case studies. In the first case, we validate the quality and accuracy of the method by comparing the stereo reconstruction of a stratocumulus layer with reflectivity observations measured by a cloud radar and the cloud base height estimated from a Lidar-ceilometer. The second case analyzes a rapid cumulus evolution in the presence of strong wind shear.

## 1 Introduction

Ground-based photogrammetry has a large potential to complement cloud observations from classical remote sensing via radiometers, radars and lidar due to its high spatial and temporal resolution. So-called hemispheric sky imagers provide a complete hemispheric view of the cloudy sky at arbitrary time intervals. Up to now, such imagers are predominantly used only for the derivation of cloud cover or cloud type classification. The derivation of additional information related to cloud size and extension including their temporal development, especially of convective boundary layer clouds, would provide valuable information for radiation closure studies under cloudy conditions and can be used for validation of LES-scale cloud simulations, e.g. from the new ICON model (Zängl et al. (2015)).

Current ground-based cloud observations are made primarily with cloud radars, lidars, lidar-ceilometers and infrared and microwave radiometers, all of which usually only sense clouds along a pencil beam; they record the 3D cloud evolution at time resolutions during which clouds already change significantly. For instance, a cross-section scan of a cloud radar takes up to one minute with a beam width of about 0.6°; moreover its sensitivity does not allow to detect the cloud boundaries. A lidar-ceilometer observes the cloud base height with high temporal resolution, but only as zenith point-measurement. Recent works show that stereo photogrammetry may help to close this gap to some extent due to the capability of cameras to capture the visible parts of clouds instantaneously with a high spatial and temporal resolution. The resulting 3D cloud geometries can then be combined with the observations from other instruments to provide valuable information for cloud reconstruction.

In this paper we investigate the potential to derive the 3D morphology of clouds with two hemispheric sky imagers, cf. to Figure 1. Fisheye cameras provide a large field of view, have robust mechanics and are very cost effective. Two cameras with a spatial displacement and simultaneous time of exposure provide the necessary information for a 3D reconstruction within an area of about $10 \times 10$ km$^2$ around the cameras. Such stereo techniques are a well studied field in photogrammetry and computer vision and early approaches of cloud photogrammetry date back to the late 19th century (Koppe, 1896).

3D stereo reconstruction is based on triangulation. Knowing the two camera orientations and the direction vector (baseline) between the cameras, each pair of corresponding image points can be back-projected into ray directions which intersect in the mapped 3D point. This requires accurately known parameters for the interior orientation, e. g. focal length and lens distortion, and also accurate knowledge of the exterior orientation, namely the position and orientation of the cameras in space, both of which need to be determined by a calibration procedure.

The main contribution of this paper is an approach to combine the large field of view of a fisheye camera with an efficient out-of-a-box dense stereo matching algorithm in order to obtain consistent and detailed cloud geometries above the area around the cameras. We achieve this by employing an epipolar rectification technique on the recorded images that is designed for fisheye cameras and is required to apply the dense stereo correspondence algorithm used in this study. Although epipolar rectification is not required for a dense reconstruction in principle, many dense stereo algorithms require rectified images because computation is greatly simplified. In contrast to regular feature-based methods used in previous studies on cloud photogrammetry, dense methods seek a correspondence for every pixel in the stereo images, leading to a dense 3D point cloud. At the same time dense stereo methods often impose spatial consistency constraints, which allows us to obtain more reliable correspondences in low-contrast image regions, which are typical for clouds, than sparse feature-based methods. A more complete and consistent cloud shape can be used in radiative transfer applications where cloud geometry is modeled explicitly. Cloud evolution studies can benefit from the larger geometric data basis regarding segmentation and classification of individual clouds, tracking and visualization, making further analysis more effective. Once the system is calibrated our approach runs fully automated and provides dense 3D geometries over large parts of the hemisphere observed by the fisheye cameras.

The paper is organized as follows. In Sec. 2, we discuss previous studies and their contributions to the field of cloud photogrammetry. In Sec. 3, we describe the fisheye camera model, the applied camera calibration techniques, as well as the epipolar rectification method and the triangulation. Sec. 4 introduces the employed dense stereo algorithm to obtain corresponding image points. Sec. 5 presents our stereo setup, a geometric uncertainty analysis and two stereo reconstruction case studies. One

case shows a reconstructed stratocumulus layer which serves as a validation for the achieved geometric accuracy including comparisons with lidar-ceilometer and cloud radar observations. The second case analyzes the cloud development under strong convection and wind shear and illustrates the quality of the cloud morphology reconstruction.

## 2 Related Work

Recent studies employed cameras with wide-angle lenses in a stereo setup to recover 3D information of clouds using feature-based methods. Seiz (2003) exploited a stereo pair of consumer cameras with an 800 m baseline to derive the cloud base height within a field of view of about $100°$ with errors well below 5 %. Hu et al. (2010) used a stereo camera system with a spatial offset of 1.5 km oriented towards the Santa Catarina Mountains near Tuscon, Arizona, to observe the diurnal cycle of orographic convection in three dimensions. Recently, Öktem et al. (2014) used a stereo camera setup for the observation of maritime clouds near Biscayne Bay, Florida, with a baseline of 873 m; their results show a good agreement with heights obtained with a lidar, yielding errors of mostly below 2 % for shallow clouds and up to 8 % for high cirrocumulus clouds. They also compared the derived cloud motion of the individual cloud layers with wind speed measurements from radiosondes. In Öktem et al. (2014) they extend their approach to marine convection. The studies of Seiz (2003) and Öktem et al. (2014) show that an accurate cloud reconstruction is possible with a stereo camera system. To the best of our knowledge, only Hu et al. (2010) and Romps and Öktem (2015) used stereo vision to reconstruct a convective cloud.

Experiments involving sky imagers focused on the derivation of the cloud base height. Allmen and Kegelmeyer (1996) used two Whole-Sky-Imagers (WSI) to derive cloud base heights; a standard ordinary stereo matching method failed due to the rather large distance between the cameras of 5 km. More than 50 % of the estimated cloud heights deviated by less than 5 % from heights obtained by the lidar-ceilometer. Kassianov et al. (2010) compared cloud base heights derived from two virtual and two real fisheye cameras located at the ARM site in the Southern Great Planes, with a baseline of 540 m. They used stochastic simulations to create a virtual cloud field and used the virtual fisheye projections for stereo vision. Comparisons with micropulse lidar observations showed that typical errors were about 10 % for low-level clouds up to 2 km high. Recently, Nguyen and Kleissl (2014) used a plane-sweep-like approach with a baseline of 1230 m between the sky imagers. Although a plane-sweep technique is also capable to produce a dense 3D geometry (Häne et al. (2014)), their implementation assumes a horizontal cloud field without any vertical structures and is aimed at computing the cloud base height for short-term solar radiation forecasting. They also implemented a correlation-based approach similar to Allmen and Kegelmeyer (1996) that operates on the unrectified fisheye images and can be extended to a global approach including spatial consistency constraints.

In our work, we use two hemispheric sky imagers in a stereo setup with a baseline of 300 m. In contrast to previous studies, we use an efficient dense out-of-the-box stereo method to recover dense 3D cloud geometries (Figure 1). Dense stereo methods obtain more geometric information than feature-based methods, especially in image regions with low contrast, which is a general problem in cloud photogrammetry. This additional geometric information can prove beneficial in cloud evolution and radiation closure studies where the cloud geometry is modeled explicitly. We evaluate our results by comparing them with cloud base height observations from a lidar-ceilometer and reflectivity profiles of a cloud radar.

## 3 Camera Calibration and Stereo Reconstruction

In this section, we describe the projection model for fisheye cameras and we formulate the geometric relationship between two sky imagers, which is required for 3D stereo reconstruction. We also introduce an epipolar image rectification scheme for fisheye stereo cameras that allows to identify corresponding image points between the two images using a dense stereo matching algorithm. Finally, we describe the employed camera calibration methods and the triangulation of 3D points using corresponding image points in the epipolar rectified images.

### 3.1 Interior Orientation of a Fisheye Camera

The interior orientation of a camera describes the camera specific projection of light onto the image plane. In a central projection each ray passes through the projection center in one particular direction. A camera is calibrated when its calibration parameters are known and this direction can be computed for every image point.

The camera model contains a projection function, which should be close to the projection of light in the optics. The interior orientation consists of the camera calibration parameters of this model, describing the camera specific projection on the image plane. The projection can be split up into a mapping of a 3D point $\mathbf{P}$ to a 2D point $\mathbf{x}'$ on the model image plane, and a mapping of $\mathbf{x}'$ to $\mathbf{x}$ to the actual pixel coordinates on the sensor plane (Figure 2). While most cameras follow the pinhole camera model (Sonka et al., 1999; Stockman and Shapiro, 2001), fisheye cameras have lenses with a different projection function and follow the omnidirectional camera model (Kannala and Brandt, 2006; Bakstein and Pajdla, 2002), visualized in Figure 2. Each projection ray passes through the projection center $C$ and intersects the image sphere in the point $\mathbf{x}''$, which determines the ray direction. The *optical axis* intersects the image plane in the *principal point* $\mathbf{x_C}$.

$\mathbf{x}''$ can be mapped to $\mathbf{x}'$ on the image plane using e.g. one of used projection functions $r(\theta)$ provided by Abraham and Förstner (2005). Each symmetric projection function $r(\theta)$ defines the distance between $\mathbf{x}'$ and the principal point $\mathbf{x_C}$ as a function of the zenith angle $\theta$ between the incoming projection ray and the optical axis as depicted in Figure 2 (a). Accordingly, the coordinates of $\mathbf{x}'$ on the image plane are a function of the azimuth angle $\varphi$ and $r(\theta)$ and are given by

$$\mathbf{x}' = \begin{bmatrix} \cos(\varphi)r(\theta) \\ \sin(\varphi)r(\theta) \end{bmatrix}.$$

The mapping of $\mathbf{x}'$ in Cartesian image coordinates to $\mathbf{x}$ in pixel coordinates is usually described as an affine transformation

$$\mathbf{x} = \begin{bmatrix} u \\ v \end{bmatrix} = \begin{bmatrix} c & 0 & u_0 \\ 0 & c & v_0 \end{bmatrix} \begin{bmatrix} \mathbf{x}' \\ 1 \end{bmatrix}$$

which depends on the camera constant $c$ and the principal point $\mathbf{x_0} = (u_0, v_0)^\top$, i.e. the principal point $\mathbf{x_C}$ in pixel coordinates. Note that the origin of the sensor coordinate system lies in the upper left corner of the image as depicted in Figure 2 (b).

Due to lens imperfections real camera projections do not follow a projection model perfectly. The lenses of the camera might be shielded by an additional glass dome as in our setup which additionally refracts the light before it enters the lens. Radial symmetric distortions result in either a barrel or pillow like stretching or bending of the image with increasing distance from

the principal point. Such distortions can be compensated by adding even-powered polynomials to the radial distance function following Brown (1971)

$$\triangle u = L(\hat{r})\hat{u} = A_1\,\hat{u}\,\hat{r}^2 + A_2\,\hat{u}\,\hat{r}^4 + A_3\,\hat{u}\,\hat{r}^6$$

$$\triangle v = L(\hat{r})\hat{v} = A_1\,\hat{v}\,\hat{r}^2 + A_2\,\hat{v}\,\hat{r}^4 + A_3\,\hat{v}\,\hat{r}^6$$

with $\hat{u} = u - u_0$, $\hat{v} = v - v_0$ and $\hat{r} = \sqrt{\hat{u}^2 + \hat{v}^2}$. $A_1$, $A_2$ and $A_3$ denote the respective coefficients of the polynomial.

In summary, we formulate the mapping into (distorted) image point coordinates $\tilde{\mathbf{x}} = (\tilde{u}, \tilde{v})^\top$ on the sensor plane as

$$\begin{aligned}
\tilde{u} &= u + \triangle u = c\,\cos\varphi\,r(\theta) + u_0 + \triangle u \\
\tilde{v} &= v + \triangle v = c\,\sin\varphi\,r(\theta) + v_0 + \triangle v
\end{aligned} \tag{1}$$

The reverse mapping of a distortion-corrected image point $\mathbf{x}$ in pixel coordinates to the 3D direction vector $\mathbf{x}''$ with unit length is given by

$$\begin{bmatrix} x \\ y \\ z \end{bmatrix} = \begin{bmatrix} \cos\varphi\sin\theta \\ \sin\varphi\sin\theta \\ \cos\theta \end{bmatrix} = \begin{bmatrix} \frac{x'}{r}\sin r \\ \frac{y'}{r}\sin r \\ \cos r \end{bmatrix}$$

where $x'$ and $y'$ are normalized image coordinates

$$x' = \frac{u - u_0}{c} \qquad y' = \frac{v - v_0}{c}$$

and $r = \sqrt{x'^2 + y'^2}$ the respective value of the radial projection function $r(\theta)$. The equidistant projection $r(\theta) = \theta$ fits the projection of our sky imagers best. Sec. 3.5 and Sec. 3.6 describe the calibration procedure to determine the parameters of interior orientation with distortion parameters as well as the calibration of a stereo camera pair.

## 3.2 Exterior Orientation and Epipolar Geometry

The omnidirectional camera model refers to the local camera coordinate system with the projection center as the origin and the sensor plane defining its orientation. The exterior orientation of a camera, which is described by three rotation angles and three translation shifts, is described in a common world reference system $\Omega_W$ and allows to derive the geometric relationship between two or more cameras. We choose one camera as the reference camera, which is considered as the left camera and the other as the right camera, which simplifies the following notation and avoids misconceptions. The choice of the reference camera has no impact on the reconstruction results.

Figure 3 illustrates the principal stereo configuration with two hemispheric cameras, making the world reference system $\Omega_W$ and the two camera reference systems $\Omega_L$ and $\Omega_R$ explicit. Let $\mathbf{C_L}$ be the world coordinates of the left camera and $\mathbf{P_L}$ an object point in the left camera reference frame. The transformation of $\mathbf{P_L}$ into world coordinates reads as

$$\mathbf{P} = (R_L\,\mathbf{P_L}) + \mathbf{C_L} \tag{2}$$

with the rotation matrix $R_L = R_x(\alpha_L)\,R_y(\beta_L)\,R_z(\gamma_L)$ and $\mathbf{C_L} \in \mathbb{R}^3$. Here $\alpha_L$, $\beta_L$ and $\gamma_L$ are the Eulerian angles (roll, pitch, yaw) and $R_x(\alpha_L)$, $R_y(\beta_L)$ and $R_z(\gamma_L)$ the respective rotation matrices. Considering $R_L$ and $R_R$ as the rotation matrices and $\mathbf{C_L}$ and $\mathbf{C_R}$ as the world coordinates of the left and right camera, we obtain the relative orientation between the left and the right camera via a rotation matrix $R$ and the baseline vector $\mathbf{t}$ via

$$R = R_L^\top R_R \qquad \text{and} \qquad \mathbf{t} = R_L^\top \left( \mathbf{C_R} - \mathbf{C_L} \right). \tag{3}$$

The determination of an accurate relative pose is crucial, as errors in the estimated exterior orientations may sum up to larger errors in the relative orientation which compromises the image correspondence algorithm and the triangulation of 3D point coordinates as investigated by Hirschmüller and Gehrig (2009).

The two camera centers $\mathbf{C_L}$, $\mathbf{C_R}$ and the object point $\mathbf{P}$ span the *epipolar plane*. This geometry can be expressed with the coplanarity equation and holds when

$$\mathbf{x_L''}^\top E\ \mathbf{x_R''} = 0 \qquad \text{with} \qquad E = [\mathbf{t}]_\times\, R, \tag{4}$$

where $E$ is the essential matrix obtained by a matrix multiplication of $R$ with the skew symmetric matrix $[\mathbf{t}]_\times$ of $\mathbf{t}$.

Given two direction rays $\mathbf{x_L''}$ and $\mathbf{x_R''}$ of corresponding image points $\mathbf{x_L}$ and $\mathbf{x_R}$ in the left and the right camera images, $E^\top \mathbf{x_L''}$ defines the normal vector of the epipolar plane spanned by $\mathbf{x_L''}$ and $\mathbf{t}$, and hence requires its correspondence $\mathbf{x_R''}$ to lie on the intersection circle between the epipolar plane and the image sphere. The same holds for $E\mathbf{x_R''}$ and $\mathbf{x_L''}$. In case of deviations from this constraint, Eq. (4) defines the cosine of the angle between the two epipolar planes spanned by $\mathbf{t}$ together with $E^\top \mathbf{x_L''}$ and $E\mathbf{x_R''}$ respectively, which we are using as an error measure to estimate $E$ using image point correspondences: The essential matrix $E$ can be expressed with five independent parameters, two for the baseline direction $\mathbf{t}$ and three for the rotation angles. We use an adapted version of the direct method of Longuet-Higgins (1981) to compute $E$ with eight correspondences, which exploits that the eigenvalues of $E$ are $\lambda_1 = \lambda_2 = 1$ and that we use spherically normalized ray directions.

The estimated essential matrix $E$ can be decomposed to obtain the relative rotation $R$ and baseline vector $\mathbf{t}$ (Hartley and Zisserman (2003)), which can be used for triangulation and epipolar image rectification, as described in Sec. 3.3. However, a 3D reconstruction based on $R$ and $\mathbf{t}$ alone takes place only in the coordinate system of the reference camera and only up to scale because of the scale ambiguity. A meaningful reconstruction in world coordinates requires the absolute length of the baseline (distance between the cameras) and the absolute orientation of the reference camera. Sec. 3.5 and Sec. 3.6 present methods to estimate both, the parameters of exterior and interior orientation.

### 3.3 Epipolar Rectification

Once the epipolar geometry and the interior orientation is known, the input images can be transformed in such a way that corresponding image points lie on the same image row, which reduces the search for corresponding image points from two dimensions (image) to one (image row). In the frame of pinhole-type cameras, epipolar image rectification refers to the computation and application of a homography which maps epipolar lines (projections of epipolar planes on the image plane) to image rows. In the omnidirectional camera model however, epipolar lines become epipolar curves due to the non-linear projection

and thus cannot be mapped by a homography because of its line-preserving character. Therefore, we employ the rectification scheme following Abraham and Förstner (2005) which is sketched in Figure 4. The epipolar rectification allows to rectify a fisheye image over a broad spectrum of the angle $\theta$, which allows to use the complete image content of a fisheye image, which is not possible via perspective rectification. However, epipolar rectification leads to lower accuracies at the margins as the image is stretched in these areas, cf. to Schneider et al. (2016).

The following derivations with respect to $\beta$ and $\psi$ are only valid for an epipolar rectified image pair. For each real camera we can define a virtual camera (subscript $_V$), such that the virtual cameras are in a canonical stereo setup, i.e. both have a common $x$-axis, the same orientation ( $R_{L,V} = R_{R,V} = I$ ) and are only shifted along the virtual $x$-axis $\mathbf{t}_V = (\|\mathbf{t}\|, 0, 0)^\top$. An object point $\mathbf{P_V}$ in the virtual world coordinate system is then defined by the angle $\beta$ which denotes the respective epipolar plane, and the two angles $\psi_L$ and $\psi_R$ that define the angle of the projection ray within the epipolar plane (Figure 4 (b)). Based on this geometry, we are able to define a rectification scheme (Figure 4 (d)) , which covers the whole 3D space: the image rows correspond to the angle $\beta$, which represents the orientation of the epipolar plane, while the image columns represent the respective angles $\psi_L$ and $\psi_R$ in the rotated epipolar plane

$$\mathbf{x}'_V = \begin{bmatrix} \psi \\ \beta \end{bmatrix} = \begin{bmatrix} \text{atan2}(y_V, z_V) \\ \text{atan2}(x_V, \sqrt{y_V^2 + z_V^2}) \end{bmatrix} \quad \text{with} \quad \mathbf{x}''_V = \begin{bmatrix} x_V \\ y_V \\ z_V \end{bmatrix}, \quad \text{atan2}(z_V, y_V) = \begin{cases} \arctan \frac{z_V}{y_V} & \text{for } y_V > 0, \\ \text{sgn}(z_V) \cdot \frac{\pi}{2} & \text{for } y_V = 0, \\ \arctan \frac{z_V}{y_V} + \pi & \text{for } y_V < 0 \wedge y_V \geq 0, \\ \arctan \frac{z_V}{y_V} - \pi & \text{for } y_V < 0 \wedge y_V < 0 \end{cases}$$

where $\mathbf{x}''_V$ corresponds to a projection ray within the frame of a virtual camera.

Let $\mathbf{x}''_L$ and $\mathbf{x}''_R$ be the projection rays of an object point $\mathbf{P}$ in the left and the right camera coordinate system ($\Omega_L$, $\Omega_R$) respectively, and $\mathbf{x}''_{L,V}$ and $\mathbf{x}''_{R,V}$ the corresponding projection rays in the virtual coordinate systems ($\Omega_{L,V}$, $\Omega_{R,V}$). The mapping between the real and virtual coordinate system follows a two-step procedure: In the first step, $\mathbf{x}''_L$ and $\mathbf{x}''_R$ are mapped from the local camera coordinate systems ($\Omega_L$, $\Omega_R$) to the world coordinate system $\Omega_W$. If we do not have knowledge about the world coordinate system $\Omega_W$, we choose $\Omega_W = \Omega_L$. From the essential matrix $E$, we extract the rotations $R_L = I$ and $R_R = R$, which map from camera coordinates to world coordinates. This leads to an equal coordinate system orientation, see step 1 in Figure 4 (a). In the second step, we construct an appropriate rotation matrix $R_V$ in order to align each systems $x$-axis with the baseline $\mathbf{t}$, see step 2 in Figure 4 (a).

Since the matrix columns of $R_V$ are the images of the base vectors $\mathbf{e_i}$, the first column is the normalized baseline vector. We can freely choose the other two coordinate axes as long as they form an orthonormal system, because each realization aligns the $x$-axis with the baseline. This means, that the rectification scheme is defined up to a rotation about the baseline $\mathbf{t}$, which corresponds to a shift of the range of the angle $\beta$ and a vertical translation in the rectified image. We define the virtual $y$-axis in the $x$-$y$-plane of the world coordinate system, which also determines the virtual $z$-axis.

Thus we finally get

$$R_V^{-1} = [\mathbf{e_1}, \mathbf{e_2}, \mathbf{e_3}] \qquad \text{with} \qquad \begin{aligned} \mathbf{e_1} &= \mathbf{t} \|\mathbf{t}\|^{-1}, \\ \mathbf{e_2} &= (-y_T, x_T, 0)^\top \, \| (-y_T, x_T, 0) \|^{-1}, \\ \mathbf{e_3} &= \mathbf{e_1} \times \mathbf{e_2}. \end{aligned}$$

Given the angular information $\beta$, $\psi_L$ and $\psi_R$ as well as the camera constant $c$, we get the rectified image coordinates by

$$\begin{aligned} u_V &= c\,\psi + u_{0,V} \\ v_V &= c\,\beta + v_{0,V} \end{aligned} \qquad \text{with} \qquad u_{0,V} = c\,\pi/2 \quad \text{and} \quad v_{0,V} = c\,\pi/2.$$

The reverse mapping, from rectified image coordinates to world coordinates is given by

$$\begin{bmatrix} x \\ y \\ z \end{bmatrix} = R_V \begin{bmatrix} \sin(\psi) \\ \cos(\psi)\sin(\beta) \\ \cos(\psi)\cos(\beta) \end{bmatrix} \qquad \text{where} \qquad \begin{aligned} \psi &= (u_V - u_{0,V})/c \\ \beta &= (v_V - v_{0,V})/c. \end{aligned} \tag{5}$$

### 3.4 Triangulation for 3D-Reconstruction

Having corresponding image points $\mathbf{x_L}$ and $\mathbf{x_R}$ identified in the image rows of the epipolar rectified images, $\mathbf{x_L''}$ and $\mathbf{x_R''}$ can be directly derived from $\mathbf{x_{L,V}}$ and $\mathbf{x_{R,V}}$ using the reverse mapping of the rectification scheme of Eq. (5). Due to the rectification the ray directions are guaranteed to lie in the 3D epipolar plane and do intersect. Considering the geometry shown in Figure 5 we identify the relation $s \cdot \sin(\psi_R + \frac{\pi}{2}) = b \cdot \sin(\gamma)$. With $\gamma = \psi_L - \psi_R$ and $\mathbf{P} = s \cdot \mathbf{x_L''}$ we have

$$\mathbf{P} = b \left( \frac{\cos(\psi_R)}{\sin(\psi_L - \psi_R)} \right) \mathbf{x_L''}. \tag{6}$$

### 3.5 Parameters of the Interior Orientation

For the estimation of the parameters of the interior orientation in Eq. (1) we employ a test field with markers that encode a geometric relationship. Such a test field can be a sophisticated setup in a laboratory (Seiz, 2003) or - as in our case - a pattern printed or fixed on a plane or inside an open half-cube as depicted in Figure 6. The calibration generally proceeds in two steps: The first step provides sample images of the pattern in different poses covering the field of view. For each image an image processing routine detects and extracts the image coordinates of the pattern geometry. In the second step, the extracted image points are used to estimate the optimal parameters of the camera model with an adjustment procedure.

We employ a software developed by Abraham and Hau (1997), that accepts input images of a calibration cube with a fixed white dotted pattern. Each inner cube side has a fingerprint pattern to make sure the detected dots are properly identified as lying on the x-, y- or z-plane, which determines their corresponding absolute 3D coordinates with respect to the cube reference system. The extraction stage results in a set of correspondences between 2D image points and 3D cube points, which are then used in a nonlinear bundle adjustment that iteratively minimizes the reprojection error between the observed image points and the reprojections of the 3D points of the pattern using the respective parameter estimation according to Eq. (1) and (2).

### 3.6 Parameters of the Exterior Orientation

First, we describe how to estimate the absolute location and orientation of each camera in the world reference system. This information is then used to derive a first estimate of the essential matrix $E^*$, which will then be iteratively refined using point-feature correspondences obtained from the stereo images according to the epipolar constraint in Eq. (4).

Employing a satellite navigation system like GPS allows us to derive the geographic position of the cameras with an accuracy of about 2–3 m. Accuracies in the range of centimeters can be achieved by using additional correction information broadcasted by terrestrial reference stations (D-GPS). The obtained coordinates can be mapped from the global reference system, e.g. WGS-84, to a local reference system using a suitable projection in order to get the exact baseline length and direction.

A more challenging task is the estimation of the camera orientation. Hu et al. (2010) uses geographic landmarks with known coordinates, Öktem et al. (2014) use the horizon and Seiz (2003) exploits stars as geometric references. As Seiz, we use sensed stars in the images, see Figure 7, as observations to estimate the absolute camera orientations. This requires a set of reference stars which can be observed by the cameras in the local night sky. The coordinates of the stars can be obtained from a star catalog like Stellarium or from online sources, e.g. of the NAOJ[1]. The coordinates are usually provided with the (north-aligned) azimuth angle $\varphi_n$ and altitude angle $\theta$ and have to be converted to 3D unit vectors according to

$$\mathbf{x_s} = \begin{bmatrix} \varphi_n \\ \theta \end{bmatrix} \quad \longrightarrow \quad \mathbf{x}''_{s} = \begin{bmatrix} \cos(\varphi_n + \pi/2)\cos(\theta) \\ -\sin(\varphi_n + \pi/2)\cos(\theta) \\ \sin(\theta) \end{bmatrix}.$$

In order to get the coordinates of the each reference star in the recorded image, we first take several long-exposure night sky images, compute the median image and subtract the median image from the original night sky images. As a result, only the moving stars are left and we can compute the respective centroid coordinates. The correct identification of the stars in the image is currently done manually by adjusting the rotation angles $\alpha$, $\beta$ and $\gamma$ until the projections are close enough to the centroids to be attributed. After the conversion of the stars image coordinates and catalog coordinates to 3D unit vectors ($x''_s$ and $x''_{cat}$), they are used to estimate the rotation $R_{abs}$ of the camera via a Levenberg-Marquardt minimization (Madsen et al., 2004) of the angular error

$$\arg\min_{R_{abs}} \quad \left\{ \sum_{i \in stars} (1 - (\mathbf{x''_{cat}}^\top (R_{abs}\,\mathbf{x''_s})))^2 \right\},$$

where $R_{abs}$ can be parametrized as a unit quaternion or as axis-angle representation.

From the absolute location and orientation of the cameras we can derive the relative orientation using Eq. (3) and a first estimate $E^*$ of the essential matrix can be composed according to Eq. (4).

For a further refinement of the essential matrix $E^*$, we collect SIFT-point-feature correspondences (Lowe (2004)) that are consistent with the epipolar constraint in Eq. (4): For each detected feature in the left image, we can select all features in the right image that are consistent with $E^*$ up to a predefined error threshold, e.g. $\angle(E^{*\top}\mathbf{x''_L}, E^*\mathbf{x''_R}) < 2°$, and then find the best

---

[1] http://eco.mtk.nao.ac.jp/cgi-bin/koyomi/cande/horizontal_rhip_en.cgi, last accessed April 2016.

match via Nearest-Neighbor using the SIFT feature descriptor. The same is done in the other direction, i.e. from the right to the left image, so that only mutually consistent matches are selected.

A couple of image pairs are enough to collect plenty of evenly distributed correspondences, as is shown in Figure 8. Since this set of correspondences will contain mismatches that would lead to a flawed refinement of $E^*$, we use the robust parameter estimation technique RANSAC (Fischler and Bolles, 1981) to filter out those likely mismatches.

Finally, we employ Levenberg-Marquardt minimization of the cost function

$$\arg\min_E \quad \left\{ \sum_{i \in \text{inliers}} \sin^2\left(\angle(\mathbf{x}_L'', \hat{\mathbf{x}}_L'')\right) + \sin^2\left(\angle(\mathbf{x}_R'', \hat{\mathbf{x}}_R'')\right) \right\}. \tag{7}$$

Because the observations $\mathbf{x}_L''$ and $\mathbf{x}_R''$ are always subject to measurement errors, they will not lie exactly within an epipolar plane. $\hat{\mathbf{x}}_L''$ and $\hat{\mathbf{x}}_R''$ denote the estimated true locations of $\mathbf{x}_L''$ and $\mathbf{x}_R''$ that do lie exactly within an epipolar plane and are closest to the observations in an angular sense. As the estimations and the observations are unit vectors and lie on the image sphere, Eq. (7) formulates a meaningful angular error measure and its minimization provides an optimal maximum likelihood solution, see Oliensis (2002).

## 4  Stereo Matching

To calculate the 3D information of a point $\mathbf{P}$, we need to know the coordinates of the projected point on both images planes. Only if such correspondences are known, its 3D location can be computed. The aim of stereo matching algorithms is to compute such correspondence information.

The visual appearance of a scene point $\mathbf{P}$ in each camera determines if stereo matching is successful or not. Automatic stereo matching is likely to fail if there are occlusions, specular reflections, varying illumination or large scale and pose differences between the images, so that either corresponding object points are not visible in both images or differ significantly in their appearance with respect to shape and size. Also objects may lack sufficient texture or contrast, or a unique surface does not exists that has a consistent visual appearance when observed from different perspectives. Especially the latter poses a problem in cloud photogrammetry. Hence, depending on the cloud situation stereo reconstruction has limitations. In practice, one either aims at finding the correspondences between distinct points in the images (sparse stereo) or between all pixels (dense stereo). A good overview is given in Scharstein and Szeliski (2002). We only employ sparse stereo during the estimation of the essential matrix (Figure 8) as described in Sec. 3.6.

Dense stereo can be advantageous when dealing with complex and dynamic scenes that have limited texture, because it effectively delivers reasonable results for image regions with low-contrast. It propagates information from high-contrast regions into the low-contrasts regions assuming similar depth at nearby pixels with similar intensity. In such regions local methods may deliver few or no information leading to a sparse point cloud, which makes further analysis like segmentation or classification difficult.

To simplify the search for correspondences, dense methods usually require epipolar rectified images, see Sec. 3.3. As a result of that, corresponding pixels are restricted to lie on the same image row, which reduces the search space from 2D to 1D.

The correspondence information is stored in the so-called *disparity map* $D$, that contains for each pixel in the rectified reference image the horizontal sub-pixel distance $d$ to its corresponding image point shifted in the same row in the other image, see Figure 9. Hence, for the two corresponding points $\mathbf{x_{L,V}}$ in the left and $\mathbf{x_{R,V}}$ in the right image, we have for each pixel position in the disparity map $D(\mathbf{x_{L,V}}) = |u_{L,V} - u_{R,V}|$ and therefore have the relation $\mathbf{x_{R,V}} = (u_{R,V}, v_{R,V})^\top = (u_{L,V} - d, u_{R,V})^\top$.

In our current approach, we rely on a dense matching algorithm that is based on the Semi-Global Matching (SGM) proposed by Hirschmüller (2005) and is called Semi-Global Block-Matching (SGBM). It produces accurate results while being deterministic and computationally efficient. In this work we use the implementation provided in OpenCV.

For a detailed algorithmic description, we refer to the original paper by Hirschmüller (2005) and to the OpenCV documentation. Here, we present only a short summary.

The basic problem of finding an optimal disparity map can be formalized as an energy minimization problem involving an energy functional and an appropriate minimization technique (Scharstein and Szeliski, 2002). A good disparity map should satisfy at least the following two aspects:

1. Two corresponding pixels should have similar intensity or structural values (data consistency).

2. Neighboring pixels with similar intensity or structural values should have similar disparity values (smoothness assumption).

Both aspects can be combined to form the global energy $E(D) = E_{data} + E_{smooth}$ and are realized in SGBM as follows: To achieve data consistency (aspect 1), $E_{data}(\mathbf{x})$ is computed for each pixel $\mathbf{x}$ independently using a window-based similarity measure such as sum of absolute/squared differences or normalized cross-correlation (NCC), yielding one matching cost per pixel for each valid disparity value $d \in [d_{min}, d_{max}]$. Note, that using a larger window will smooth the disparity map since small details have a smaller influence on the measure. This causes fine structures to disappear, but also reduces errors caused by image noise. However, relying only on a minimum in $E_{data}$ will cause mismatches, especially in regions with low- contrast or repeating patterns, which results in a flawed disparity map. This can be countered by introducing an additional smoothness term $E_{smooth}$, that penalizes larger disparity differences of neighboring image points across eight linear paths.

Figure 9 illustrates the computation of $E_{data}$ using epipolar rectified images and the final disparity map after incorporating $E_{smooth}$. One can clearly see the difference in the disparity values between clouds that are closer (high disparity) and more distant clouds (smaller disparity).

In our application, we use a window-size of $11 \times 11$ pixels. To achieve a successful matching in larger low-contrast regions and reduce the variability in the reconstruction due to the noisy image signal, we scale the input images to one quarter size. This causes an oversmoothing near cloud boundaries, but this way we obtain smoother cloud boundary.

## 5 Stereo Setup and Results

In this section we present our stereo setup deployed at the Forschungszentrum Jülich GmbH, Germany. We also give a geometric uncertainty analysis of the current setup, discuss common error sources and how they affect the calibration and reconstruction results with a focus on asynchronous recordings. Finally, we present our experimental results of two case studies. The first case shows that our dense stereo approach is able to achieve a geometric accuracy that is comparable with those of previous studies using sparse stereo methods, like Seiz (2003) and Öktem et al. (2014). The second case illustrates the capability of our approach to successfully reconstruct the complex 3D cloud structure and dynamics of convective clouds.

### 5.1 Camera Setup

We exploit two sky imagers installed at the Forschungszentrum Jülich GmbH, Germany, which also hosts the Jülich Observatory for Cloud Evolution (JOYCE Löhnert et al. (2014)), which has been developed in the framework of the Transregional Collaborative Research Center TR32 (Simmer et al. (2015)). For the evaluation of our results we use observations from a local lidar-ceilometer and cloud radar (Figure 10). The first camera is located at 50.90849° N, 6.41342° E and the second at 50.90613° N, 6.41144° E, resulting in a baseline length of approximately 300 m. Compared to previous studies that mention a baseline between 500 m (Kassianov et al. (2010)) and 900 m (Öktem et al. (2014)), this is a rather short distance and results in higher geometric uncertainty of the estimated 3D points on the clouds, but reduces occlusions and enhances the ratio of mutually visible cloud regions in both images. Furthermore, the short baseline increases the similarity of the cloud appearance in both images, which is beneficial for stereo matching. A more in-depth analysis of these aspects is presented in the next section.

Both cameras are IDS network-cameras of type uEye GigE UI-2280SE with a 2/3" CCD sensor consisting of 2448×2048 pixels and are equipped with a Fujinon FE185C057HA-1 C-Mount Fisheye adapter, providing a 185° field of view and fixed focus. The cameras are mounted in a box and point towards the sky. An acrylic glass dome protects the cameras against environmental effects. A power supply and a fan distribute heat to prevent the condensation of water on the glass dome. Each camera is connected to a small computer that hosts a self-developed camera control application, based on the IDS C++ SDK (IDS, 2013) which allows us to control the cameras remotely, e.g. for scheduled recordings with settings as exposure time, recording interval (e.g. 15 seconds) or modes like long-exposure (night mode) or High Dynamic Range (HDR). The images are currently saved locally and transferred if needed. Synchronization is done by frequent requests to a local NTP service.

### 5.2 Geometric Uncertainty

First, we discuss the general spatial accuracy of a 3D reconstruction assuming correct orientation parameters, but a flawed disparity estimate. In order to understand the individual contribution of each parameter to the depth uncertainty, we use the standard formulation for pinhole cameras (Kraus (2004)), where the disparity is modeled by the parallax $p_x$ and its uncertainty $\sigma_{p_x}$. The parallax is the angle between two corresponding projection rays, i.e. $\gamma$ in Figure 5, which can also be formulated as the distance between corresponding projections on the image plane (Figure 12). Given a stereo system of identical cameras

with camera constant $c$ (cf. Sec. 3.1) and baseline length $t$, as illustrated in Figure 12, we have the horizontal coordinate $X$ and the depth $D$ given by

$$X = x'_L \frac{t}{p_x} \quad D = c \frac{t}{p_x}$$

We focus on the absolute depth uncertainty $\sigma_D$. From the relation between the relative depth uncertainty $\sigma_D$ and the relative parallax uncertainty $\sigma_{p_x}$

$$\frac{\sigma_D}{D} = \frac{\sigma_{p_x}}{p_x}$$

we can formulate the depth accuracy in several ways

$$\sigma_D = \frac{D}{p_x} \sigma_{p_x} = \frac{c\,t}{p_x^2} \sigma_{p_x} = \frac{D}{c\,t/D} \sigma_{p_x}$$

The first identity simply states that the (nominal) uncertainty grows linear if the whole setup is scaled up (increasing depth and baseline), the second term indicates that the uncertainty is proportional to the squared inverse of the disparity. As a consequence, deviations at higher disparities are less significant to $\sigma_D$ than deviations at smaller disparities. An analogous statement is that at smaller angles $\gamma$ in Figure 5, deviations in $\psi_L$ or $\psi_R$ cause higher errors. The last identity shows that uncertainty is inverse to the ratio of baseline length to depth value. In other words, doubling the baseline $t$ while maintaining a fixed distance to an object will double the accuracy (increased $\gamma$).

These considerations assume that the image points can be identified and matched with a specific parallax uncertainty $\sigma_{p_x}$. However, larger baselines (or disparities) usually affect the stereo matching because increasing parts of the object might not be visible by both cameras. Additionally, the object will have significantly different geometric appearance in each camera. Thus a tradeoff between accuracy and geometric completeness and consistency is necessary to get the best results. Compared to previous studies, the small baseline of our stereo setup leads to noisy and inconsistent reconstructions beyond 5 to 6 km. However, our current focus lies on boundary layer clouds and their lateral morphology, which usually have a horizontal spacing of just a couple of kilometers between each other. Also, distant clouds are often occluded by others so that a larger baseline does not always offer the desired benefits.

Figure 11 shows the reconstruction error for a virtual cloud layer at 1500 m and 3000 m height over a $10\times10$ km$^2$ area around the cameras, assuming an error of 1 pixel in the disparity map after the matching phase (Sec. 4), which corresponds to a directional error of approximately $0.1°$. The values represent absolute errors within the epipolar plane. Therefore, depending on the horizontal distance to the cameras, the error has a larger vertical component (small distance) or a larger horizontal component (larger distance). The error grows larger with increasing distance to the cameras and with increasing co-linearity between object point and the camera centers. In both cases, the angle $\gamma$ between the projection rays becomes very small, yielding larger triangulation errors. Thus, sky imagers do not provide hemispheric 3D reconstruction with homogeneous accuracy. However, this deficit can be ameliorated by employing a third camera in a triangle configuration. A successful integration of a third camera into the dense stereo matching scheme including epipolar rectification is explained for perspective cameras in Heinrichs and Rodehorst (2006). An adaption to omnidirectional cameras has, to the best of our knowledge, not been addressed yet, but seems to be possible.

Next, we compare the spatial resolution of a sky imager with a wide-angle camera used in previous studies. Fisheye lenses cover a substantially larger field of view than normal perspective lenses, but at an reduced effective angular resolution. As a consequence, the stereo depth resolution is lower for fisheye lenses compared to perspective ones. This drawback limits the effective range for a high quality reconstruction, especially for distant clouds. A comparison of our fisheye cameras with one of the wide- angle cameras from Öktem et al. (2014) highlights the differences: While our camera has a field of view (FOV) of about $180°$ and the circular view field covers a 3.5 Megapixel region (from a 5 Megapixel sensor), the wide-angle camera has a FOV of $67°$ and takes 1-Megapixel images from a 5-Megapixel sensor. To compare the view fields, the respective solid angle $\Omega_{fish}$ for the sky imager and $\Omega_{wa}$ for the wide angle camera – given in steradians – must be derived. Assuming a field of view of $180°$ for the sky imager and $67°$ for the wide-angle camera leads to $\Omega_{fish} = 6.28$ and $\Omega_{wide} = 1.04$.

Furthermore, the solid-angle per pixel is $6.28/3.5 \cdot 10^{-6} = 1.8 \cdot 10^{-6}$ for the fisheye and $1.04/1.0 \cdot 10^{-6} = 1.04 \cdot 10^{-6}$ for the wide angle camera, resulting in a $43\%$ lower spatial resolution of the fisheye camera. Using the full resolution of the wide-angle camera with 5 Megapixel, the ratio would be $11\%$. Hence, one must use a sensor almost 10 times higher resolution to compete with the wide-angle camera in this respect. Depending on the fisheye projection function and the location in the image, the spatial resolution will of course vary due to the different degree of distortion.

The imaging process of a sensor adds a random noise signal, which can be limited, but not avoided. In principle, this also affects parameter estimation, because both localization and measurement are disturbed. Given a large number of measurements for the calibration, the signal noise can be compensated in a maximum likelihood estimation as the redundancy is high. The stereo matching is also affected by the noisy image signal and causes a disturbed 3D reconstruction.

In the following analysis we investigate the effects of an asynchronous recording of the stereo images during the observation of a dynamic cloud scene.

Despite frequent requests to an NTP service, we sometimes experience asynchronous system times on the local computers in the range of a few seconds. Consequently, the whole cloud scene shifts between the single shots and thus causes a displacement in the images, which leads to a biased or flawed disparity map. We investigate the effects of a cloud field displacement of $\Delta = \pm 15$ m along the baseline (x-direction) and perpendicular to the baseline (y-direction) in a virtual sky imager setup together with a virtual cloud layer at 3 km, using the 3D rendering software Blender (Blender Foundation, 2016). The virtual sky imagers have identical internal camera geometries comparable to the real ones, and a relative pose of $R_L = R_R = I$ and $\mathbf{t} = (300, 0, 0)^{\top}$ m. Figure 13 shows the cross-sections of the respective reconstruction along the baseline (x-axis) and perpendicular to the baseline (y-axis). A displacement along the x-axis results in a lower (2875 m for $\Delta = -15$ m) or a higher (3183 m for $\Delta = +15$ m) cloud base compared to the unaffected reconstruction (3025 m for $\Delta = 0$ m), while a displacement along the y-axis just causes an overall higher standard deviation of the reconstruction without a systematic error in the mean base height ($\sigma(\Delta = +15$ m$) = 45$ m, $\sigma(\Delta = -15$ m$) = 48$ m and $\sigma(\Delta = 0$ m$) = 35$ m). The results confirm that a displacement in x-direction is equivalent to a change in the length of the baseline $\mathbf{t}$. Hence, the reconstructed cloud base $\hat{h}$ compared to the real cloud base $h$ can be derived according to $\hat{h} = h \cdot [\, \|\mathbf{t}\|/(\|\mathbf{t}\| + \Delta)\,]$.

## 5.3    Evaluation of the 3D Reconstruction

The following two case studies are designed to evaluate our approach using observations from a lidar-ceilometer and a cloud radar, and to show its capability to capture the complex 3D shapes and dynamics of convective clouds.

We compare some of the reconstructions with observations of cloud base heights from a lidar-ceilometer and reflectivity measurements by a cloud radar, both deployed in the vicinity of the cameras at the JOYCE observation site. The lidar-ceilometer is a Vaisala CT25K which was operated with a range between 60 m and 7500 m and a beam diameter of $0.043°$, which corresponds to 2.25 m at 3000 m height. The cloud radar is a Metek polarimetric Doppler Radar (MIRA) operating at 35 GHz with a similar angular resolution over a range between 150 m and 15 km and a maximum sensitivity of -45 dBZ at 5 km. For our comparisons we average observations over an integration time of 0.15 seconds for each range resolved beam. A complete cross-section scan then takes approximately one minute for all elevation angles, that range between $15°$ and $165°$. For the empirical evaluation, we use the cloud base height observations from the lidar-ceilometer. Since the lidar-ceilometer offers only point measurements every 15 seconds, we use observations from a 10 minute period around the measurement time of the cameras to get a meaningful comparison. We compare the mean reconstructed height values from a near-zenith rectangular area of 9 km$^2$. The cloud radar offers a direct comparison between the reflectivity signal from an RHI cross-section scan and the respective cross-section of the 3D cloud shape derived by the stereo method. In order to cover the region of best geometric accuracy of the stereo method, we orientate the radar scans along an almost perpendicular direction to the baseline, compare Figure 10 and 11). After image acquisition, the input images are preprocessed, which includes a resampling to one quarter size and a contrast enhancement using the *Contrast Limited Adaptive Histogram Equalization* (CLAHE). For stereo matching with SGBM we use a window-size of $11 \times 11$ pixels and a cloud mask to remove some artifacts. After triangulation of the 3D point cloud, we create a cloud boundary mesh using methods from the open-source *Point Cloud Library* (Library, 2016).

### 5.3.1    Analysis of the 3D Reconstruction of Stratocumulus-layer Clouds

We present two cases with stratocumulus clouds, which we will use to evaluate our result by comparing it against observations from the lidar-ceilometer and the cloud radar.

Figure 14 shows the results from the 11th of August, 2014, at 14:12:00 UTC, right after a shower moved over the JOYCE site with a trailing stratocumulus layer, that was also captured by the cloud radar. The mean cloud-base heights from the reconstructions are 2881 m while the lidar-ceilometer observations result in 2897 m. Note also, that the cloud radar is – due to its measuring frequency in the microwaves – not as sensitive as the lidar-ceilometer or the camera to the visible outer cloud boundaries, with the consequence that especially the boundaries of a cloud might not be detectable by the radar, but by the camera or the lidar.

The second case has been recorded on the 5th of August, 2014, which is interesting for two reasons: First, a typical boundary layer evolution has been observed, starting with small to medium size cumulus convection along with a steady rise of the cloud base. Second, from early noon on the cloud scenery shifted towards stratocumulus layers, resulting in less convection and thus providing again good conditions for a comparison against observations from the lidar-ceilometer and the cloud radar

(Figure 15). Comparing the reconstruction result with the cloud radar indicates a reasonable estimation of the bases of the higher cloud layer and the cumulus cloud below it. However, differences between 100 to 150 m are noticeable and the lidar-ceilometer observations report a cloud-base mean height of 2922 m compared with the height of 2766 m derived from stereo reconstruction. See Sec. 5.3.3 for a discussion of the potential reasons for this discrepancy.

### 5.3.2 Analysis of the 3D Reconstruction of Cumulus Clouds

The 24[th] of July, 2014 showed strong convection with rapid cloud development and decline, providing excellent conditions to apply our dense stereo reconstruction and to capture their complex shapes and dynamics. Figure 16 shows a cumulus mediocris forming approximately 3 km away from the stereo camera pair. One can observe the convective updraft and the rising cumulus turret. While reaching a height of about 4000 m, the turret enters the higher wind field, resulting in a skewed shape of the cloud due to wind shear. Figure 17 shows a cross-section of the reconstructed 3D cloud boundary at 12:07:00 UTC together with the cloud base measured with the the lidar-ceilometer.

Figure 18 shows a smaller convective cloud, but also with a rather complex morphology from 11:28:00 UTC to 11:32:00 UTC on the same day. One observes how the developing convective turret covers parts of the cloud, which results from its increasingly concave shape. The temporally closest cloud base height values from the lidar-ceilometer between 12:14:38 UTC and 12:17:08 UTC report 1487 m on average. Figure 19 shows a cross-section of the reconstruction similar as in Figure 17.

### 5.3.3 Discussion of the Experimental Results

Our cloud reconstructions show an overall good agreement with the cloud base height observations from the lidar-ceilometer and the cloud radar. Mean near-zenith cloud base heights for the stratocumulus cases are within 1 % for the 11[th] of August and 5 % for the 5[th] of August of the lidar-ceilometer mean values, and the stereo method is able to capture the geometric shapes of the cloud bases as in Figure 15. A possible explanation for the deviation ocurring on the 5[th] of August is a shift between the computer system times as is described in Sec. 5.2. Although errors in the orientation parameters cannot be excluded, the relative orientation estimation shows a standard deviation of $0.04°$ which is comparably accurate. On the other hand, a bias in the computers system time could be observed at times and confirms this assumption. Based on the observations from the nearby wind-lidar, which reported a wind speed of about 5 m s$^{-1}$ in a direction almost collinear to the baseline, a time difference of about +3 seconds between the left and the right camera would lead to a scaled reconstructed cloud base of $2922\,\text{m} \cdot (302\,\text{m}/317\,\text{m}) = 2783\,\text{m}$ which is close to the actual reconstructed cloud base height of 2766 m.

The results from the 24[th] of July show that the dense stereo method is able to almost fully reconstruct the visible outer shape of a convective cloud. In both cases the concave and increasingly skewed shape of the cloud is nicely captured by the dense stereo method, as is illustrated in the cross-sections in Figure 17 and Figure 19. Also the cloud base is clearly visible and matches the lidar-ceilometer value of 1495 m quite well, considering that temporally close measurements are only available from 12:12:07 UTC to 12:17:08 UTC ranging from 1448 m to 1585 m.

# 6 Conclusions

In this paper, we investigate the reconstruction of the 3D geometry of clouds from fisheye cameras using dense stereo approaches from photogrammetry. We present a complete approach for stereo cloud photogrammetry using hemispheric sky imagers. Our approach combines calibration, epipolar rectification, and block-based correspondence search for dense fisheye stereo reconstruction for clouds. We show that cloud photogrammetry is able to compute the cloud envelope geometry and demonstrate the potential of such methods for the analysis of detailed cloud morphologies. By applying an epipolar rectification together with a dense (semi-)global stereo matching algorithm, we are able to compute clouds shapes that are more complete and contiguous than reconstructions relying on regular feature-based methods. Once the cameras are calibrated, the method can be fully automated to deliver real-time information of the cloud scene.

The proposed technique requires accurate camera calibration parameters and synchronously triggered cameras. Although the validation of our results with cloud radar observations should be extended to convective clouds, the reconstructions have shown to be stable over time, yielding robust cloud base and motion estimates. The validation for stratiform clouds show acceptable deviations from the lidar-ceilometer and radar measurements.

The system will be permanently installed at JOYCE and record cloud evolution on an automated basis, which will provide a large data basis for more extended analysis. We will add further camera pairs at larger distances from JOYCE to enable the reconstruction of more complete cloud boundaries. Future work will also focus on the combination of cloud photogrammetry with different sensors.

*Acknowledgements.* This study has been performed within the framework of the High Definition Clouds and Precipitation for advancing Climate Prediction (HD(CP)$^2$) project funded by the Bundesministerium für Bildung und Forschung (BMBF), FKZ 01LK1212B. We want to thank Dr. Emiliano Orlandi and Dr. Kerstin Ebell from the University of Cologne for their efforts to provide the observation data from the cloud radar at JOYCE, and Dr. Birger Bohn from the Forschungszentrum Jülich GmbH for organization and maintenance. We also acknowledge the use of the infrastructure, assistance and funding by the Transregional Collaborative Research Center TR32, funded by the Deutsche Forschungsgemeinschaft (DFG).

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

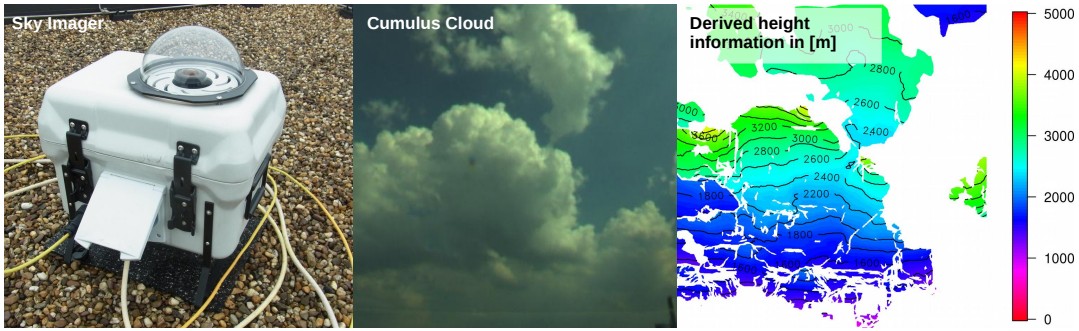

**Figure 1.** We employ two hemispheric sky imagers (left) in a stereo setup with a baseline of 300 m to derive dense and detailed geometric information such as height and morphology of clouds within a range of about 5 km using the simultaneously recorded image pairs. The derived heights (right) of an exemplary recorded cloud (middle) using the fisheye images, as shown in Figure 4 (c), of the two sky imagers.

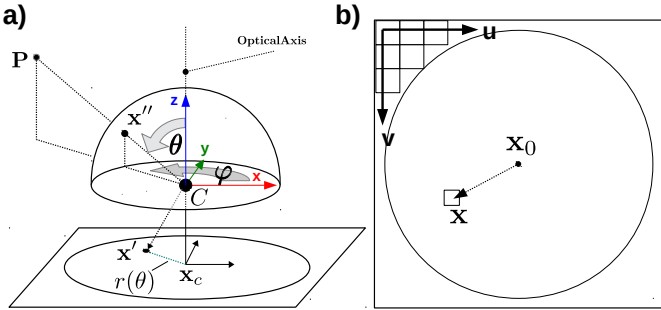

**Figure 2.** (a) In the omnidirectional camera model the 3D object point $\mathbf{P}$ is mapped to $\mathbf{x}'$ on the image plane. Several radial-symmetric projection functions $r(\theta)$ can be used, which define the distance to the projection center $\mathbf{x_C}$; (b) The final projection $\mathbf{x}$ in pixel coordinates is determined by the camera calibration parameters and additional distortion coefficients.

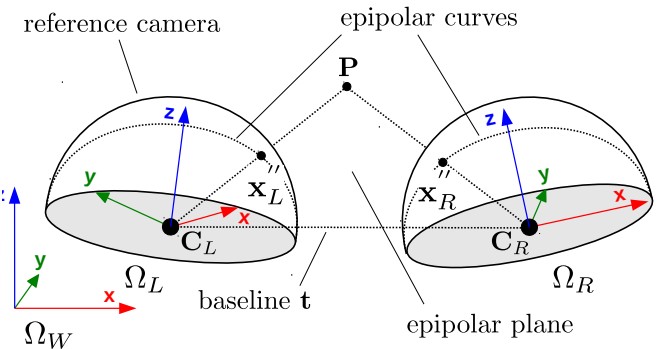

**Figure 3.** The two hemispheric cameras are located at $\mathbf{C_L}$ and $\mathbf{C_R}$ with independent orientation in the world coordinate system $\Omega_W$. The projections $\mathbf{x_L''}$ and $\mathbf{x_R''}$ of a 3D point $\mathbf{P}$ on the image hemisphere in each camera system $\Omega_L$ and $\Omega_R$ span together with the baseline $\mathbf{t}$ the epipolar plane and can be used to reconstruct $\mathbf{P}$ via triangulation.

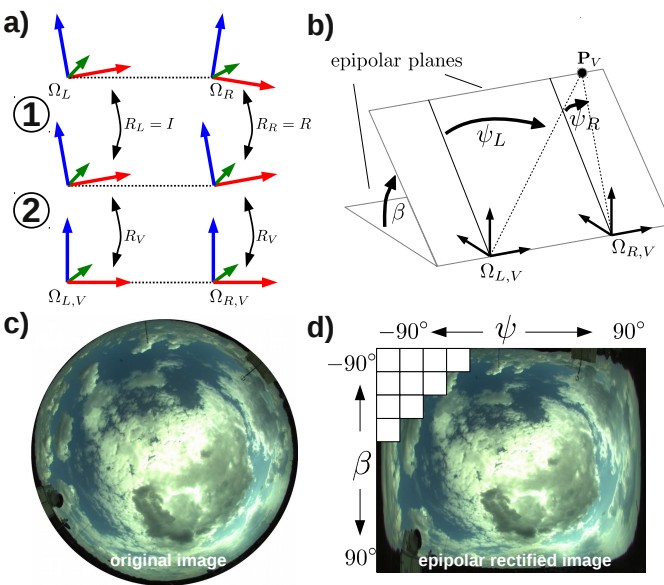

**Figure 4.** Epipolar rectification for omnidirectional cameras: The two-step rotational mapping between real and virtual cameras (a) results in a canonical camera setup of virtual cameras during rectification (b). A fisheye image (c) is rectified such that the angles $\beta$ and $\psi$ correspond to the lines and rows of the image (d).

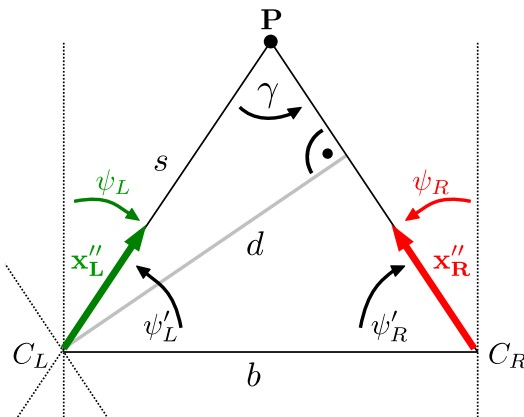

**Figure 5.** Two corresponding ray directions $\mathbf{x}_\mathbf{L}^{''}$ and $\mathbf{x}_\mathbf{R}^{''}$ are defined by the angles $\psi_L$ and $\psi_R$ within the epipolar plane. With baseline length $b$ the distance $s$ between left camera and 3D Point $\mathbf{P}$ can be derived, which allows to determine 3D point coordinates $\mathbf{P} = s\,\mathbf{x}_\mathbf{L}^{''}$.

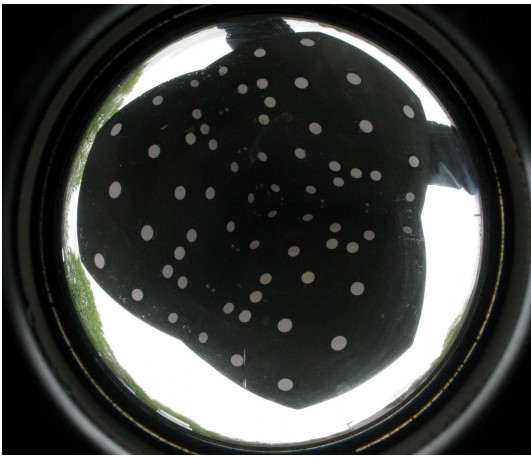

**Figure 6.** Camera calibration with cube: the pattern on the inside of the cube defines a set of 3D points with respect to the cube coordinate system and can be used as reference data to solve for the fisheye projection parameters.

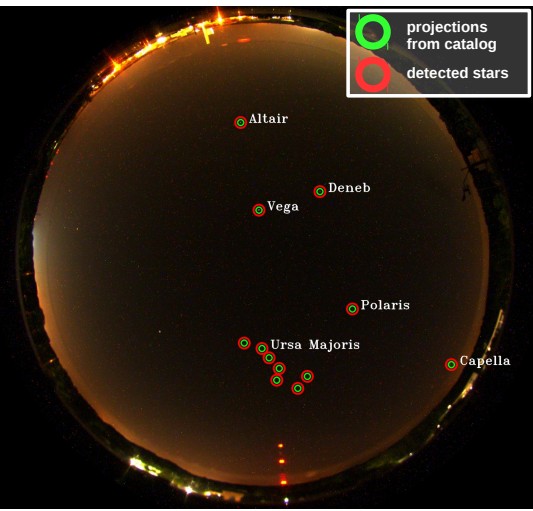

**Figure 7.** Absolute orientation estimation via stars from long-exposure images. In case of an accurate orientation estimation, the projected coordinates from the star catalog should match the detected stars in the image.

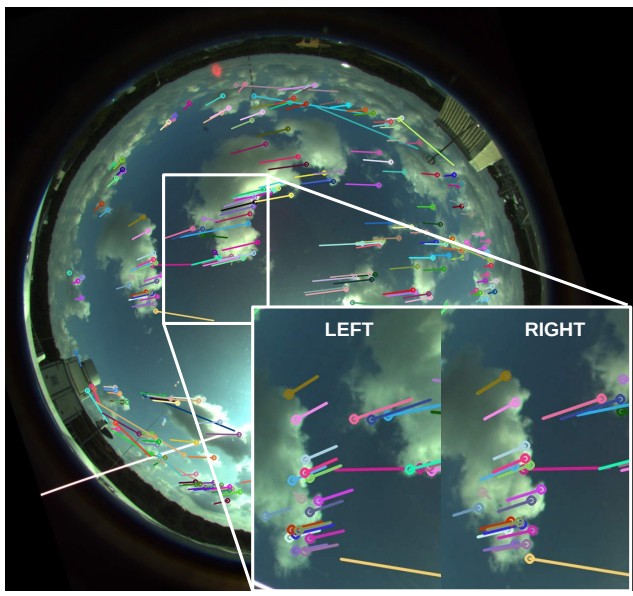

**Figure 8.** Detected interest points can be matched across the stereo images and are marked by the same color. At least 5 correspondences are needed to compute the relative orientation between the cameras and each also provides one 3D point.

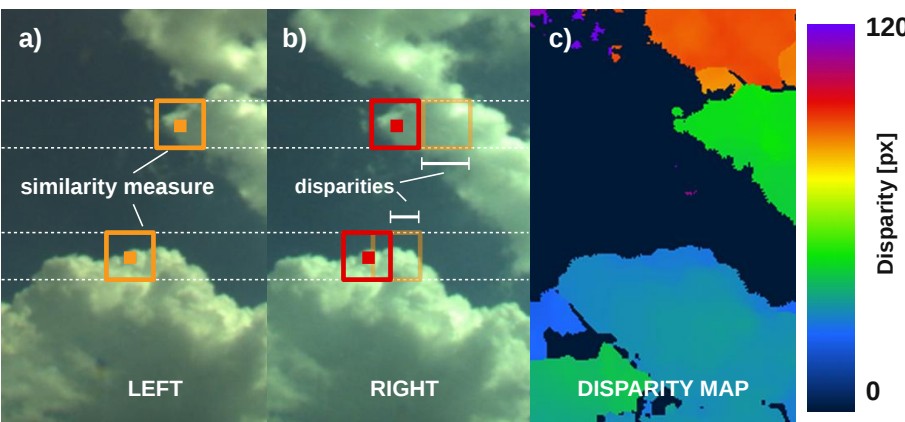

**Figure 9.** Illustration of dense stereo matching using epipolar rectified images. The correspondence information is stored in a disparity map. Each disparity $D(\mathbf{x}) > 0$ then defines a correspondence between two image points $\mathbf{x}_{L,V}$ and $\mathbf{x}_{R,V}$ and can be used for triangulation, which results in a dense 3D point cloud.

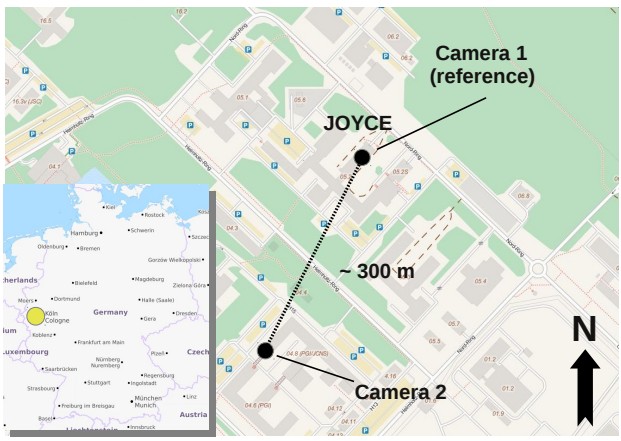

**Figure 10.** Camera setup at the Research Center Jülich (map source: *OpenStreetMap*).

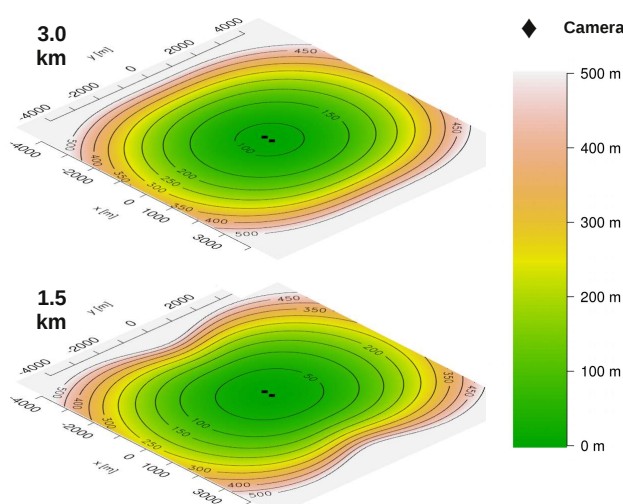

**Figure 11.** Errors in the disparity values cause a directional deviation of the projection rays within the epipolar plane. Assuming an error of $\Delta\psi_R = 0.1°$, we can compute the absolute geometric error within the epipolar plane for a hypothetical cloud layer at $1.5\,\mathrm{km}$ and $3\,\mathrm{km}$ height in an area of 5 km around the cameras.

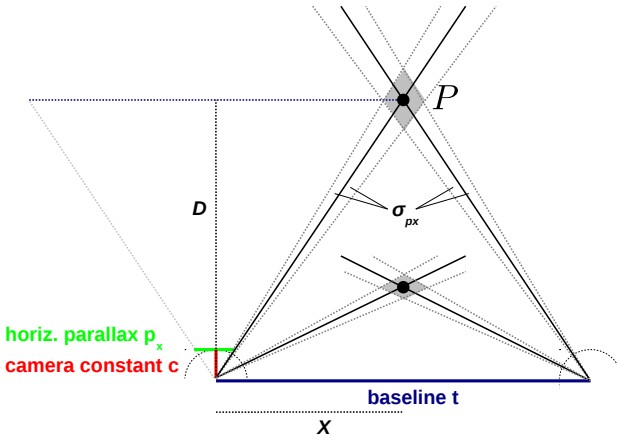

**Figure 12.** Illustration of general geometric uncertainty of the reconstruction within an epipolar plane: An uncertainty in the projection rays $\sigma_{p_x}$ introduces an uncertainty in the estimated location of $\mathbf{P}$ indicated by the gray region: A smaller/higher depth value $D$ (or higher/smaller parallax $p_x$) reduces/increases the depth uncertainty $\sigma_D$, but reduces/increases the horizontal uncertainty $\sigma_X$

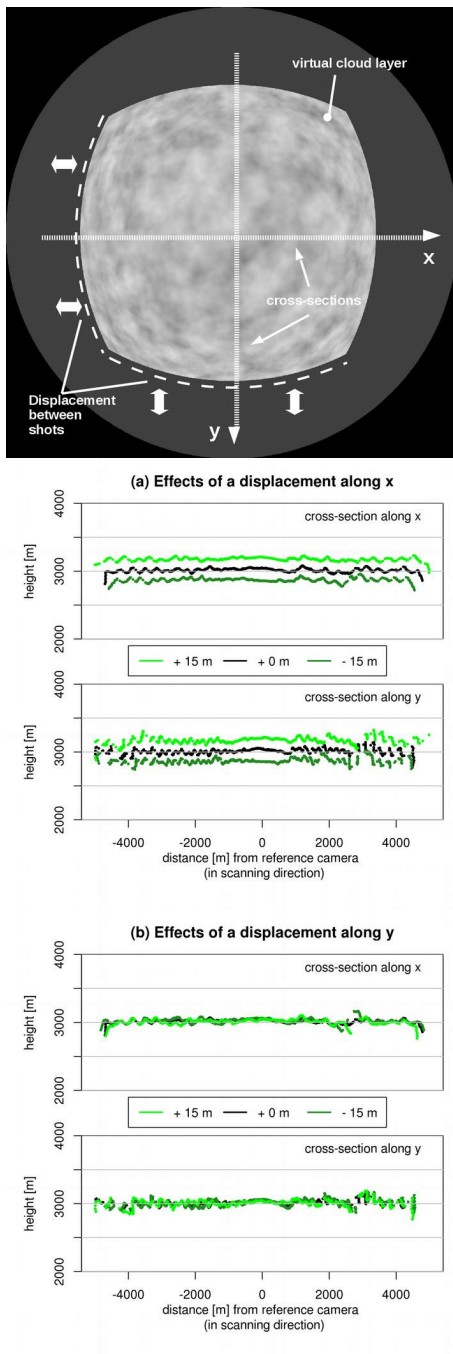

**Figure 13.** Simulation of an asynchronous recording by the stereo cameras for the same area size. A virtual cloud layer at 3 km height was displaced between the two recordings of the stereo cameras by ±15 m (a) along the baseline (x-direction) and (b) in perpendicular direction (y-axis). Plots show for each case the respective cross-sections in x- and y-direction of the reconstruction.

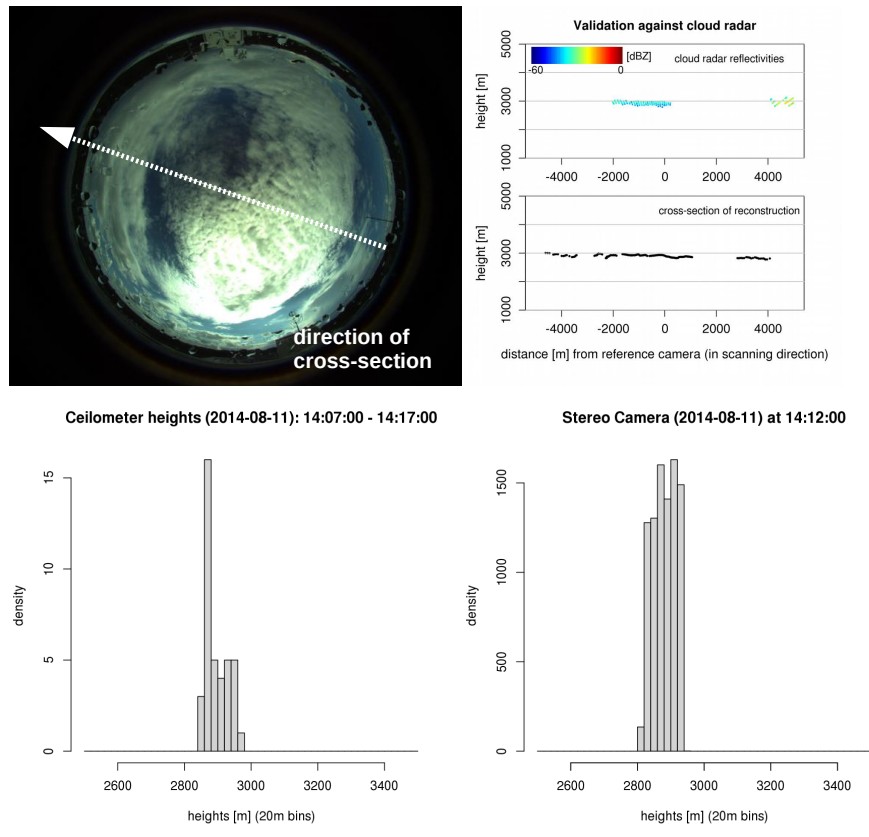

**Figure 14.** Comparison of the reconstruction with cloud radar (cross-section) and lidar (near-zenith cloud base heights) at 11 August 2014, 14:12:00 UTC. The original fisheye image showing the respective direction of the cross-sections (top left). The cross-section of the reconstruction compared with the reflectivities from the cloud radar (top right). Histogram of the cloud base heights observed by the stereo camera (bottom left) and the lidar-ceilometer (bottom right).

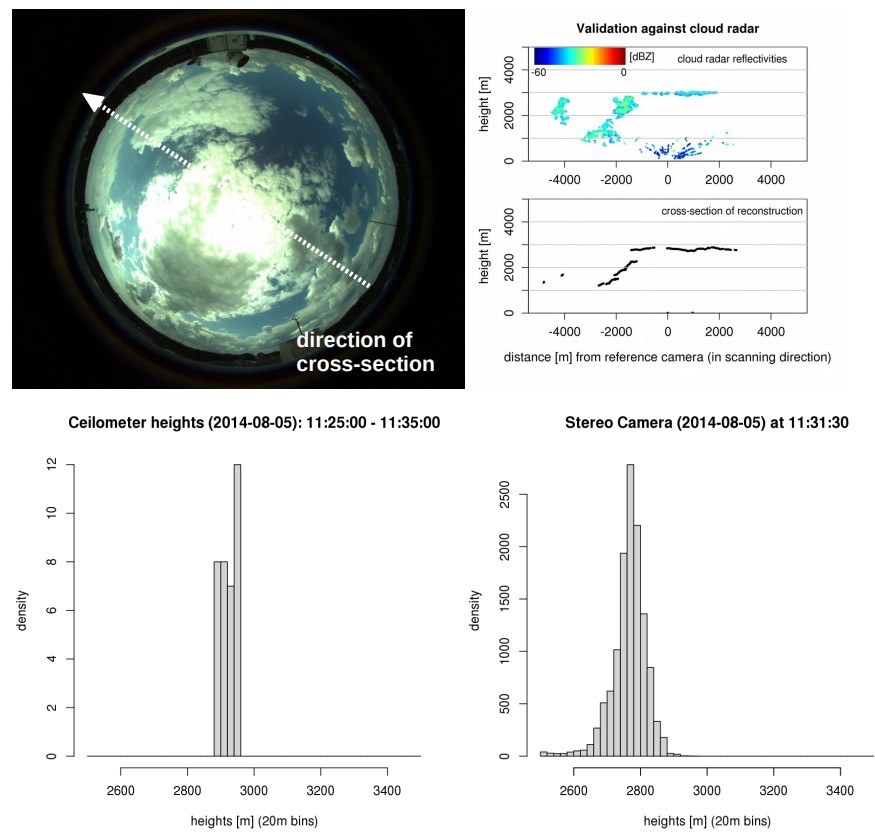

**Figure 15.** Comparison of the reconstruction with cloud radar and lidar at 5 August 2014, 11:31:30 UTC, as in Fig.14

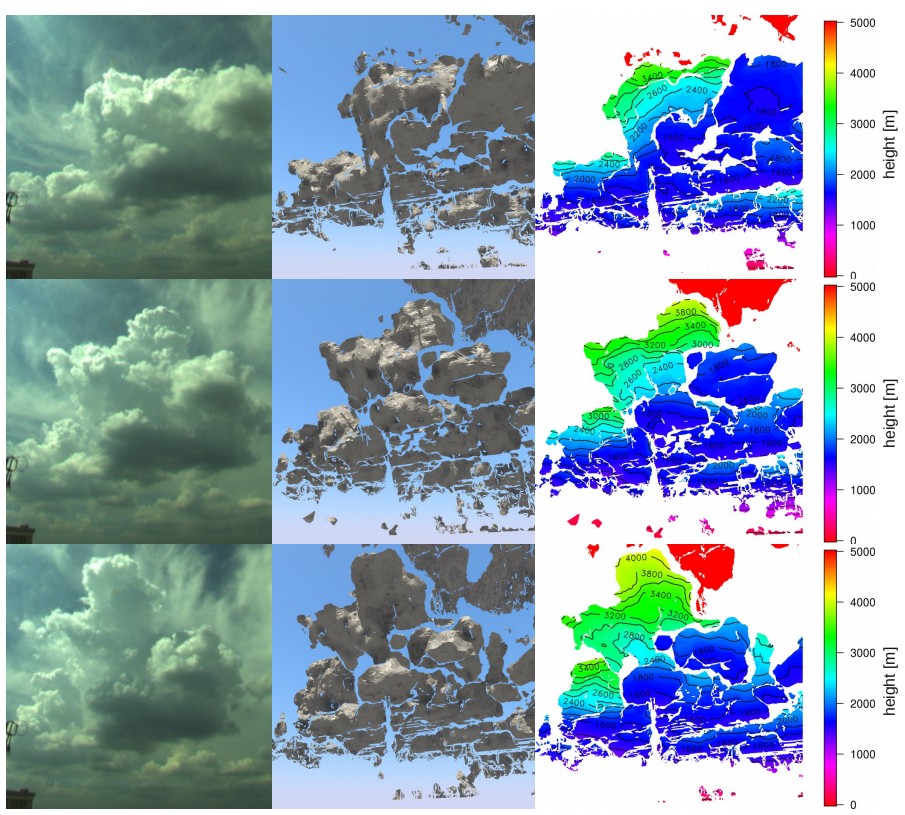

**Figure 16.** 3D reconstruction of cumulus mediodcris at 24 July 2014 approximately 3 km away: The left column shows a subsection of the images obtained from Camera 1. The central column visualizes the reconstruction as an untextured triangulated boundary mesh. The right column shows the color-coded height of the reconstruction in meters with contour lines (right). Results are shown for 12:03:00 UTC (top row), 12:05:0 UTC (middle row) and 12:07:00 UTC (bottom row).

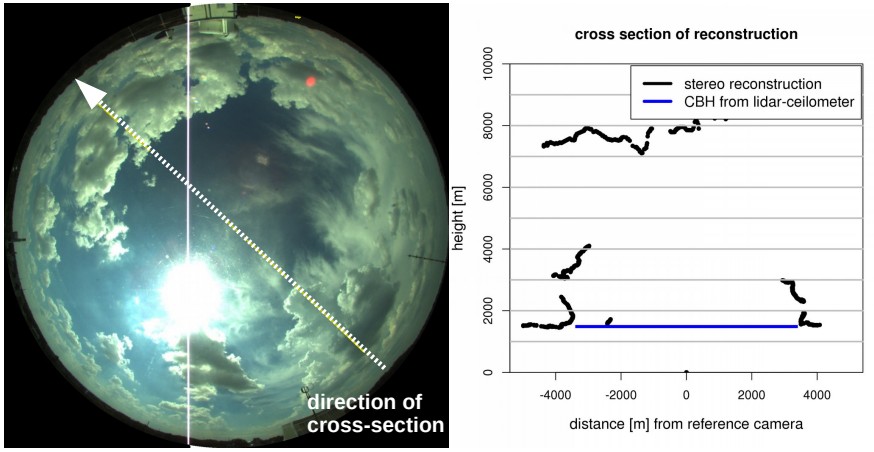

**Figure 17.** Cross-section of the reconstruction from 24 July 2014, 12:07:00 UTC, and highlighted cloud base height from the lidar-ceilometer.

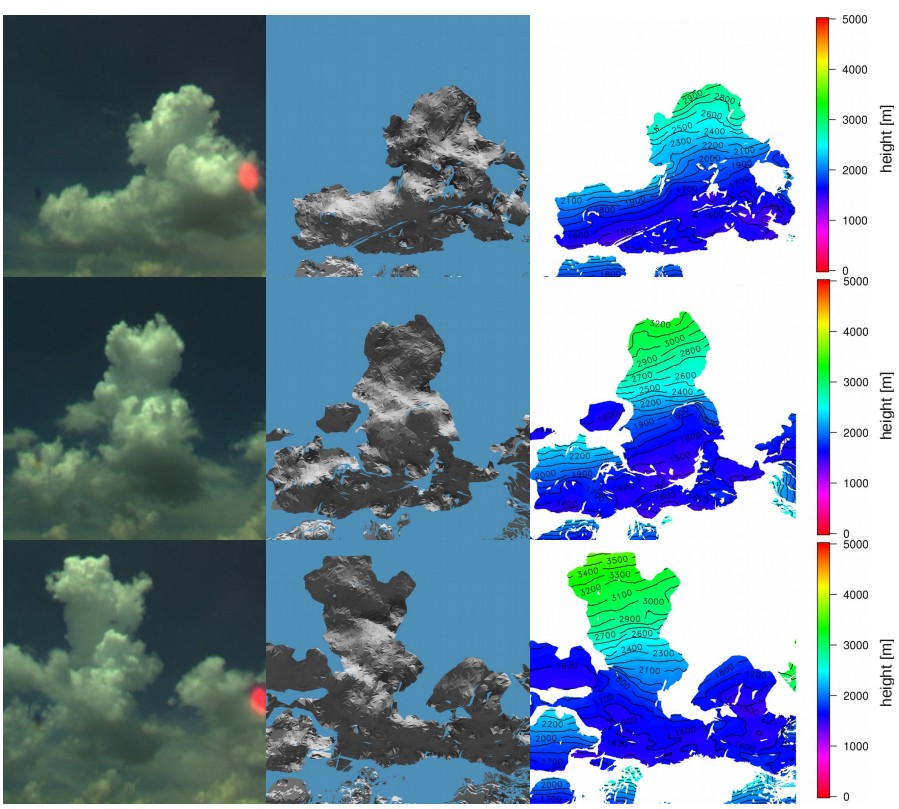

**Figure 18.** 3D reconstruction of cumulus cloud from 24 July 2014, organized as in Figure 16. Results are shown for 11:28:00 UTC (top row), 11:30:00 UTC (middle row) and 11:32:00 UTC (bottom row). The cloud is approximately 3 km away.

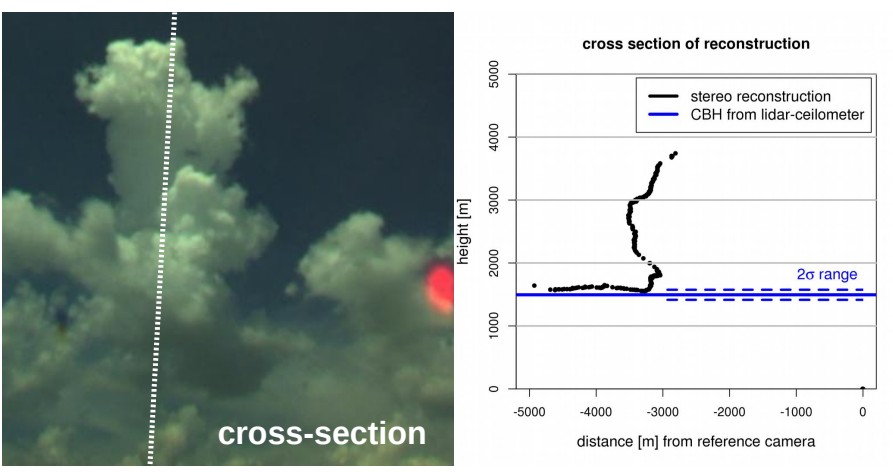

**Figure 19.** Cross-section of the reconstruction from 24 July 2014, 11:32:00 UTC.