# Peer review of "Cloud Photogrammetry with Dense Stereo for Fisheye Cameras"

_Atmospheric Chemistry and Physics, 2016_

## Referee Comment (RC1) · Anonymous Referee #1 · 9 Jun 2016

General Comments:

This manuscript presents a method to obtain dense stereo reconstruction on clouds by using a pair of cameras with fisheye lenses. Fisheye lenses provide a larger field of view but their projection model is different from pinhole models, hence they require a different calibration and rectification solution. The main contribution of this work is to combine dense stereo reconstruction methods with a stereo setup using fisheye lenses and exploit dense matching to derive 3D geometry of clouds. It presents the potential of stereo photogrammetry in providing 3D cloud geometry that is not easy to achieve with other atmospheric measurement devices. Considering that calibration is a crucial part of stereophotogrammetry, the methods described in this manuscript to achieve reliable camera calibration with a fisheye lens are of significant value. I recommend that this manuscript should be published but after carefully adressing the issues I detail below.

Specific Comments:

In many parts of the manuscript, rectification is referred as the method that allows dense stereo matching. I find this misleading, because rectification is merely a transformation to translate the epipoles to infinity so that the epipolar lines are parallel in both images, hence matching algorithm is less time consuming and more straightforward to design.

Page 3, Line 3, Romps and O ktem also studied convective clouds in the following two references Romps and Oktem, Stereo photogrammetry reveals substantial drag on cloud thermals, GRL, 2015 Oktem and Romps, Observing atmospheric clouds through stereo reconstruction, Proceedings of SPIE - The International Society for Optical Engineering, March 2015

In Section 3, the parameters such as theta and phi angles are only displayed in figures but are not introduced in the text nor in the captions. There are many parameters used in the equations, it maybe a good idea to list and define them in a separate table or introduce/explain them in the text.

In Section 3.3, it is claimed that rectification allows to use the complete image content of a fisheye image. It is not clear to me why the whole content of rectified image can be used but the whole content of the non-rectified image cannot. Besides, the distortion (stretching) introduced by the rectification is likely to severely limit the use of data beyond a certain theta.

Section 4, Line 14, "Dense stereo is advantageous when dealing with complex geometries but also effectively delivers reasonable results for image regions with low-contrast". I believe that this statement needs revision to clarify the point being made. I understand the clouds are considered as complex geometries but it is not clear to me how dense stereo is advantageous for these cases.

Technical Corrections:

Abstract, Line 8, "..of the a cloud ...", "the" should be omitted.

Abstract, Line 10, there appears to be tense mismatch.

Introduction, Page 1, Line 22, "...cloud observation.." should be "observations"

Introduction, Page 2, Line 9, "..which both need to be", correct as "..both of which need to be..."

Page 3, Line 27, "..orientation consist of...", correct as "..orientation consists of..."

Page 4, equations with phi uses lowercase phi in one equation, and uppercase phi in another, do they refer to different parameters?

Page 6, the last equation which is below Line 30, Is this the intended notation for e.g. "atan2(z_v,y_v)"? I believe that the notation should be revised to clearly state the formula.

Page 9, the first equation on the top, what is "sin(delta)"?

Page 10, Line 2, "..on the both..", I recommend "...on both of the..."

Page 10, Line 10, better separate the last statement into two sentences to make it more readable.

Page 11, Line 32, "...mounted in a box with ...", is "with" redundant in this statement?

Page 12, Line 3, "..(IDS, 2013).", remove "."

Page 12, Line 8, "..an reduced..", correct as "..a reduced.."

Page 12, Line 17, "Further, the solid angle..", correct as "Furthermore, .."

Page 13, Line 9, correct "citepblender"

Page 13, Line 26, is "..0.15 seconds..." supposed to be "... 15 seconds ..."?

---

## Author Comment (AC1) · 15 Jul 2016

We thank the referee for the important and valuable comments on the paper. We will use them to correct the paper for the final version. Our response to the referees comments are as follows:

-> Referee Comment:

"In many parts of the manuscript, rectification is referred as the method that allows dense stereo matching. I find this misleading, because rectification is merely a transformation to translate the epipoles to infinity so that the epipolar lines are parallel in both images, hence matching algorithm is less time consuming and more straightforward to design."

[Figure]

Authors Response:

We agree with the referee. Our intention was to emphasize that epipolar rectification is usually a prerequisite to efficient out-of-the-box dense stereo matching techniques, like the mentioned Semi-Global-Matching approach. Also other approaches like for example the Graph-Cut technique usually require rectified images. On the other hand the plane-sweep algorithm does not and works on the original (or undistorted) images.

-> Referee Comment:

"Page 3, Line 3, Romps and Oktem also studied convective clouds in the following two references Romps and Oktem, Stereo photogrammetry reveals substantial drag on cloud thermals, GRL, 2015 Oktem and Romps, Observing atmospheric clouds through stereo reconstruction, Proceedings of SPIE - The International Society for Optical Engineering, March 2015"

Authors Response:

Our focus lies on the feasibility of the proposed method to reconstruct the cloud morphology in its full extent, while the referenced paper uses the stereo technique with the aim to answer a specific question that can also be answered with a feature-based method. Therefore, the main difference is the use of a dense versus a sparse stereo reconstruction. Further, we use automated matching, while the referenced paper mentions that the features are matching manually across the images. However, the paper needs to be added to the related work section.

-> Referee Comment:

"In Section 3, the parameters such as theta and phi angles are only displayed in figures but are not introduced in the text nor in the captions. There are many parameters used in the equations, it maybe a good idea to list and define them in a separate table or introduce/explain them in the text."

Authors Response:

[Figure]

We agree with the referee. In the text it says 'r(theta)' instead of 'theta'. The angle phi is not introduced. We will correct this.

-> Referee Comment:

"In Section 3.3, it is claimed that rectification allows to use the complete image content of a fisheye image. It is not clear to me why the whole content of rectified image can be used but the whole content of the non-rectified image cannot. Besides, the distortion (stretching) introduced by the rectification is likely to severely limit the use of data beyond a certain theta."

Authors Response:

Although the whole image content can be used theoretically, the rectification, but also the geometric conditioning of parts of the images limit its use. The first lines in this section are a bit misleading in this regard. We therefore agree with the referee and will correct this in the final version. Of course the image content of a non-rectified image can also be used, e.g. for feature-based matching. However, the matching algorithm we use requires epipolar rectified images and non-rectified images are therefore not an option. As is shown in Fig.11 a large negative impact is due to the geometric conditioning along the baseline as is also stated in Öktem et al. (2014). Since the stretching occurs mostly at the left and right sides of the image, the impact is largest in these regions. We did not experience a larger negative effect along the central meridian (top, zenith, bottom).

-> Referee Comment:

"Section 4, Line 14, "Dense stereo is advantageous when dealing with complex geometries but also effectively delivers reasonable results for image regions with low-contrast". I believe that this statement needs revision to clarify the point being made. I understand the clouds are considered as complex geometries but it is not clear to me how dense stereo is advantageous for these cases."

Authors Response:

Matching errors during a sparse feature-based reconstruction are likely to result in a quite different cloud shape because every feature is comparatively important due to their small amount. Also, features are generally matched independently across the image, without any corrective term like in a globally optimized matching. Hence, not only the likelihood of error occurrences, but also their respective impact on the final reconstruction is larger, than with global matching. Another issue is that we can simply expect more 3D information about the object, which is preferable for a later visualization, but also for processing, e.g. segmentation and tracking, since a larger data basis is provided for these tasks.

Technical Corrections: If not directly responded to, comment will be taken into account for correction.

-> Referee Comment:

"Page 4, equations with phi uses lowercase phi in one equation, and uppercase phi in another, do they refer to different parameters?"

Authors Response:

No. This is an error and will be corrected.

-> Referee Comment:

"Page 6, the last equation which is below Line 30, Is this the intended notation for e.g. "atan2(z_v,y_v)"? I believe that the notation should be revised to clearly state the formula."

Authors Response:

Yes. The full notation will be inserted in the final version.

-> Referee Comment:

"Page 9, the first equation on the top, what is "sin(delta)"?"

Authors Response:

This is an error. 'delta' should be 'theta'.

-> Referee Comment:

"Page 13, Line 26, is "..0.15 seconds..." supposed to be "... 15 seconds ..."?"

Authors Response:

This refers to the averaging time of a single range resolved scan (a single elevation value / beam), which takes 0.15 seconds. The whole scan (elevation angle from 15 degrees to 165 degrees) takes almost 1 minute.

---

## Referee Comment (RC2) · Anonymous Referee #2 · 5 Aug 2016

The paper describes 3d reconstruction of clouds from two ground-based fisheye cameras in a stereo configurations using photogrammetry and dense image matching. The results are compared in a case study with those obtained by radar and lidar instruments.

Being not an expert in the field of cloud observation, I would be interested to read a bit more about the significance of the work: what does this technique offer, compared to the existing ones? Is it a better resolution/accuracy, a larger area, reduced cost, more completeness? One could also describe it in terms of requirements: we want to see ..., but existing techniques only give ... and therefore we develop a new system expecting to get ... and here we evaluate it.

There are several challenges being addressed, concerning the "difficult" geometry of

fisheye lenses, the size of the setup (with a baseline of 300m), the use of automatic (dense) matching with "fuzzy" objects (clouds). All is well explained, but it is not always completely clear how it relates to the state of the art and where is the novelty - I suppose it is in the application of dense matching to clouds, but in that case the results could be analyzed a bit more exactly there.(Furthermore, I liked the used of stars in the exterior orientation).

Only in the evaluation section it becomes apparent what one had in mind concerning the size of the area to be measured: results are shown up to 4 km away from the cameras at two different heights (Fig 11) . The accuracy gets rather poor at larger distances, which may be due to the baseline of only 300m (but a larger one might affect matching performance). Some more discussion about this would be welcome. By the way, what is the area in other Figures , like 15 and 17?

The paper mentions using a third camera, but that would not be covered by the current setup of resampling the images to epipolar geometry (would it?). This implication should be mentioned.
* * *

---

## Author Response (AR1)

Authors Response to the reviews of

**Cloud Photogrammetry with Dense Stereo for Fisheye Cameras**

by

Christoph Beekmans et al.

**Table of Contents**

Dear Editors and Reviewers,

thank you very much for your valuable comments on our paper. We addressed all the issues raised by the reviewers and modified/extended the paper as requested. Below, you can find the answers to the questions raised by the referees.

We also want to inform you that we included Mr. Martin Lennefer into the list of co-authors due to his intensive involvement in our work regarding his technical expertise of the camera system and his contributions to the design and implementation of our field campaigns.

Thank you for your efforts,

Christoph Beekmans, Johannes Schneider, Thomas Läbe, Martin Lennefer, Cyrill Stachniss and Clemens Simmer

**Point-to-Point response to: anonymous referee #1 from June 9th, 2016**

**Referee Comment:**

*„In many parts of the manuscript, rectification is referred as the method that allows dense stereo matching. I find this misleading, because rectification is merely a transformation to translate the epipoles to infinity so that the epipolar lines are parallel in both images, hence matching algorithm is less time consuming and more straightforward to design."*

**Authors Response:**

We agree with the referee that this can be misleading. Our intention was to emphasize that epipolar rectification is usually a prerequisite to use efficient out of the box dense stereo matching techniques, like the Semi-Global-Matching employed in this paper. The plane-sweep algorithm works also on unrectified images.

To be more precise we added changes at several places.

1) In the abstract we point out, that the rectification "allows the use of efficient out-of-the-box dense matching algorithms designed for classical pinhole-type cameras ..." (Page 1, Line 3)
2) We added at Page 2, Line 21, a clarification that we are aiming to use dense stereo methods designed for perspective cameras:
   "*The main contribution of this paper is an approach to combine the large field of view of a fisheye camera with an efficient out-of-a-box dense stereo matching algorithm[...]*"
3) We added a statement at Page 2, Line 22 in order to clarify that epipolar rectification is not a principal requirement for a dense stereo reconstruction:
   "*Although epipolar rectification is not required for a dense reconstruction in principle, many dense stereo algorithms require rectified images because computation is greatly simplified.*"

**Referee Comment:**

*„Page 3, Line 3, Romps and Oktem also studied convective clouds in the following two references Romps and Oktem, Stereo photogrammetry reveals substantial drag on cloud thermals, GRL, 2015 Oktem and Romps, Observing atmospheric clouds through stereo reconstruction, Proceedings of SPIE – The International Society for Optical Engineering, March 2015"*

**Authors Response:**

Thank you for the feedback. We have fully addressed this issue and have incorporated the suggested reference by integrating them into the related work section (Page 3, Line 14):

" *To the best of our knowledge, only Hu et al. (2010) and Romps and Öktem (2015) used stereo vision to reconstruct a convective cloud.*"

*Romps, D. M. and Öktem, R.: Stereo photogrammetry reveals substantial drag on cloud thermals, Geophysical Research Letters, 42, 5051–5057, 2015.*

**Referee Comment:**

*„In Section 3, the parameters such as theta and phi angles are only displayed in figures but are not introduced in the text nor in the captions. There are many parameters used in the equations, it maybe a good idea to list and define them in a separate table or introduce/explain them in the text."*

**Authors Response:**

We agree with the referee. In the text we corrected 'r(theta)' to 'theta' (Page 4, Line 21 and 22). The angle phi and the projection function r(theta) are now introduced on Page 4, Line 20-22:

" *Each symmetric projection function $r(\theta)$ defines the distance between x' and the principal point $x_C$ as a function of the zenith angle $\theta$ between the incoming projection ray and the optical axis as depicted in Figure 2 (a). Accordingly, the*

*coordinates of x on the image plane are a function of the azimuth angle φ and r(θ) and are given by[...]"*

**Referee Comment:**

*„In Section 3.3, it is claimed that rectification allows to use the complete image content of a fisheye image. It is not clear to me why the whole content of rectified image can be used but the whole content of the non-rectified image cannot. Besides, the distortion (stretching) introduced by the rectification is likely to severely limit the use of data beyond a certain theta."*

**Authors Response:**

We agree with the referee and we have corrected this issue. The image content of a non-rectified image can also be used, e.g. for feature-based matching. However, the matching algorithm we use requires epipolar rectified images, which allow to use out-of-the-box dense and efficient stereo methods as we mention on Page 2, Line 19-20. As we have fisheye images, we have to use a special rectification model and cannot use the classical perspective rectification using a homography (Page 6, Line 30):

" *In the frame of pinhole-type cameras, epipolar image rectification refers to the computation and application of a homography which maps epipolar lines (projections of epipolar planes on the image plane) to image rows. In the omnidirectional camera model however, epipolar lines become epipolar curves due to the non-linear projection and thus cannot be mapped by a homography because of its line-preserving character. Therefore, we employ the rectification scheme following Abraham and Förstner (2005) which is sketched in Figure 4*"

The used epipolar rectification allows to keep the complete image content, but also introduces some strong distortions at the image borders (Page 7, Line 4):

"*However, epipolar rectification leads to lower accuracies at the margins as the image is stretched in these areas, cf. to Schneider et al. (2016).*"

**Referee Comment:**

*„Section 4, Line 14, "Dense stereo is advantageous when dealing with complex geometries but also effectively delivers reasonable results for image regions with low contrast". I believe that this statement needs revision to clarify the point being made. I understand the clouds are considered as complex geometries but it is not clear to me how dense stereo is advantageous for these cases.*

**Authors Response:**

We addressed this issue and revised this statement by giving more explanation in this paragraph (Page 10, Line 26-30):

*"Dense stereo can be advantageous when dealing with complex and dynamic scenes that have limited texture, because it effectively delivers reasonable results for image regions with low-contrast. It propagates information from high-contrast regions into the low-contrasts regions assuming similar depth at nearby pixels with similar intensity. In such regions local methods may deliver few or no information leading to a sparse point cloud, which makes further analysis like segmentation or classification difficult."*

**Technical Corrections**

**_Referee Technical Comment #1:_**

*Abstract, Line 8, "..of the a cloud …", "the" should be omitted.*

**Authors Response:**

Agreed.

**Referee Technical Comment #2:**

*Abstract, Line 10, there appears to be tense mismatch.*

**Authors Response:**

Agreed: "We implemented and evaluated" has been changed to present tense.

**Referee Technical Comment #3:**

*Introduction, Page 1, Line 22, "...cloud observation.." should be "observations"*

**Authors Response:**

Agreed.

**Referee Technical Comment #4:**

*Introduction, Page 2, Line 9, "..which both need to be", correct as "..both of which need to be..."*

**Authors Response:**

Agreed.

**Referee Technical Comment #5:**

*Page 3, Line 27, "..orientation consist of...", correct as "..orientation consists of..."*

**Authors Response:**

Agreed.

**Referee Technical Comment #6:**

*Page 4, equations with phi uses lowercase phi in one equation, and uppercase phi in*

*another, do they refer to different parameters?*

**Authors Response:**

Thank you for pointing to this error, which has been corrected.

**Referee Technical Comment #7:**

*Page 6, the last equation which is below Line 30, Is this the intended notation for e.g. "atan2(z_v,y_v)"? I believe that the notation should be revised to clearly state the formula.*

**Authors Response:**

Yes. The full notation has been inserted in the paper.

**Referee Technical Comment #8:**

*Page 9, the first equation on the top, what is "sin(delta)"?*

**Authors Response:**

Thank you for identifying this is error. 'delta' has been changed to 'theta'.

**Referee Technical Comment #9:**

*Page 10, Line 2, "..on the both..", I recommend "...on both of the..."*

**Authors Response:**

"..on the both.." changed to "...on both..."

**Referee Technical Comment #10:**

*Page 10, Line 10, better separate the last statement into two sentences to make it more readable.*

**Authors Response:**

Changed to:

"*Especially the latter poses a problem in cloud photogrammetry. Hence, depending on the cloud situation stereo reconstruction has limitations.*"

**Referee Technical Comment #11:**

*Page 11, Line 32, "...mounted in a box with ...", is "with" redundant in this statement?,*

**Authors Response:**

We removed the "with".

**Referee Technical Comment #12:**

*Page 12, Line 3, "..(IDS, 2013).", remove "."*

**Authors Response:**

Agreed.

**Referee Technical Comment #13:**

*Page 12, Line 8, "..an reduced..", correct as "..a reduced.."*

**Authors Response:**

Agreed.

**Referee Technical Comment #14:**

*Page 12, Line 17, "Further, the solid angle..", correct as "Furthermore, .."*

**Authors Response:**

Agreed.

**Referee Technical Comment #15:**

*Page 13, Line 9, correct "citepblender"*

**Authors Response:**

Corrected: "*(Blender Foundation, 2016)*"

**Referee Technical Comment #16:**

*Page 13, Line 26, is "..0.15 seconds..." supposed to be "... 15 seconds ..."*

**Authors Response:**

This refers to the averaging time of a single range resolved scan (a single elevation value / beam), which takes 0.15 seconds. The whole scan (elevation angle from 15 degrees to 165 degrees) takes almost 1 minute. We added a sentence to clarify this (Page 15, Line 8-10):

"*For our comparisons we average observations over an integration time of 0.15 seconds for each range resolved beam. A complete cross-section scan then takes approximately one minute for all elevation angles, that range between 15 degrees and 165 degrees.*"

**Point-to-Point response to: anonymous referee #2 from August 5th, 2016**

We thank the referee for the important and valuable comments on the paper. We used them to correct the paper for the revised version. Our response to the referees comments are as follows:

**Referee Comment:**

"*Being not an expert in the field of cloud observation, I would be interested to read a bit more about the significance of the work: what does this technique offer, compared to the existing ones? Is it a better resolution/accuracy, a larger area, reduced cost, more completeness? One could also describe it in terms of requirements: we want to see ..., but existing techniques only give ... and therefore we develop a new system expecting to get ... and here we evaluate it*"

**Authors Response:**

The problem is that the mentioned instruments simply do not have the spatial and/or temporal resolution for near-instantaneous scans and thus representations of e.g. boundary layer clouds which are highly dynamic and comparatively small are quite limited. In case of a cloud radar they also lack sensitivity to cover all parts of a cloud.

In order to better motivate our contribution, we reformulated the respective paragraph in the Introduction (Page 2, Line 1-5):

"*Current ground-based cloud observations are made primarily with cloud radars, lidars, lidar-ceilometers and infrared and microwave radiometers, all of which usually only sense clouds along a pencil beam; they record the 3D cloud evolution at time resolutions during which clouds already change significantly. For instance, a cross-section scan of a cloud radar takes up to one minute with a beam width of about 0.6 degrees; moreover its sensitivity does not not allow*

*to detect the cloud boundaries. A lidar-ceilometer observes the cloud base height with high temporal resolution, but only as zenith point-measurement.*"

and Page 1, Line 27:

"*A more complete and consistent cloud shape can be used in radiative transfer applications where cloud geometry is modeled explicitly. Cloud evolution studies can benefit from the larger geometric data basis regarding segmentation and classification of individual clouds, tracking and visualization, making further analysis more effective.*"

Regarding the novelty and contribution of our approach in the field of cloud photogrammetry, we state in Page 2, Lines 17-24:

"*The main contribution of this paper is an approach to combine the large field of view of a fisheye camera with an efficient out-of-a-box dense stereo matching algorithm in order to obtain consistent and detailed cloud geometries above the area around the cameras.[...]. In contrast to regular feature-based methods used in previous studies on cloud photogrammetry, dense methods seek a correspondence for every pixel in the stereo images, leading to a dense 3D point cloud. At the same time dense stereo methods often impose spatial consistency constraints, which allows us to obtain more reliable correspondences in low-contrast image regions, which are typical for clouds, than sparse feature-based methods*"

We also state on Page 3, Line 16, that up to now, fisheye cameras have only been used to derive cloud base height, but not for a recovery of complete 3D cloud geometries of e.g. convective boundary layer clouds:

"*Experiments involving sky imagers focused on the derivation of the cloud base height.*"

**Referee Comment:**

"*There are several challenges being addressed, concerning the „difficult" geometry of fisheye lenses, the size of the setup (with a baseline of 300m), the use of automatic (dense) matching with „fuzzy" objects (clouds). All is well*

*explained, but it is not always completely clear how it relates to the state of the art and where is the novelty – I suppose it is in the application of dense matching to clouds, but in that case the results could be analyzed a bit more exactly there.(Furthermore, I liked the used of stars in the exterior orientation).*"

**Authors Response:**

We updated some passages in section 2 and 5 in order to put our methods and decisions better in the context of the cited methods.

The general contribution and novelty of this paper is mentioned in the introduction, which has also undergone some changes (Page 2, Lines 19-31). We now also state on Page 3, Line 27-31:

" *In contrast to previous studies, we used a dense stereo method to recover a dense 3D cloud geometry (Figure 1). Dense stereo methods obtain much more geometric information than feature-based methods, especially in image regions with low contrast, which is a general problem in cloud photogrammetry. This additional geometric information can prove beneficial in cloud evolution and radiation closure studies where the cloud geometry is modeled explicitly.*"

A state-of-the-art regarding the stereo matching depends primarily on the reconstruction application (clouds in our case) and the number of cameras involved. Multi-view reconstruction for example offers many more constraints on the scene geometry than two cameras and makes visibility and radiometric consistency assumptions more feasible. In this field, feature-based methods (at least as a starting point) are the best choice because each feature / point can be refined towards a consistent maximum-likelidhood estimate.

A two-camera stereo application has only the simple epipolar constraint and hence - assuming a relatively small baseline compared to the whole 3D scene - further constraints do not have the same significance than in multi-view applications with a variety of viewports.

Another aspect is, that clouds themselves are fuzzy objects and thus do not have clearly defined 3D boundaries that can be refined the same way as solid object boundaries can be. These implications cause a shift of our focus from geometric accuracy towards geometric completeness, although we do not think that there is a significant trade-off (compare statement about feature-based methods and dense stereo methods in section 4 or in the introduction).

**Referee Comment:**

"*Only in the evaluation section it becomes apparent what one had in mind concerning the size of the area to be measured: results are shown up to 4 km away from the cameras at two different heights (Fig 11) .* "

**Authors Response:**

We updated the abstract and the introduction as well as the caption of Figure 1 to solve this deficit:

"*We present a novel approach for dense 3D cloud reconstruction using two hemispheric sky imagers with fisheye lenses in a stereo setup above 10×10 km2.*"

and

"*Two cameras with a spatial displacement and simultaneous time of exposure provide the necessary information for a 3D reconstruction within an area of about 10×10 km2 around the cameras.*"

**Referee Comment:**

"*The accuracy gets rather poor at larger distances, which may be due to the baseline of only 300m (but a larger one might affect matching performance). Some more discussion about this would be welcome.*"

**Authors Response:**

We revised section 5.2 extensively and discussed the implications of a larger / shorter baseline or different distances of the object on the reconstruction

uncertainty and the matching performance. For example, on Page 13, Line 16, we state now:

"*However, larger baselines (or disparities) usually affect the stereo matching because increasing parts of the object might not be visible by both cameras. Additionally, the object will have significantly different geometric appearance in each camera.*"

We also want to refer to our statement in section 5.1 where we also discuss this issue at Page 12, Line 13-16:

"*Compared to previous studies that mention a baseline between 500 m (Allmen and Kegelmeyer (1996)) and 900 m (Öktem et al. (2014)), this is a rather short distance and results in higher geometric uncertainty of the estimated 3D points on the clouds, but reduces occlusions and enhances the ratio of mutually visible cloud regions in both images*"

**Referee Comment:**

"*By the way, what is the area in other Figures , like 15 and 17?*"

**Authors Response:**

Since the baseline is constant the main area of reconstruction does not change.

We added the distance information to the figure captions, e.g. for Fig. 16:

"*3D reconstruction of cumulus mediodcris at 24 July 2014 approximately 3 km away [...]*"

**Referee Comment:**

**"***The paper mentions using a third camera, but that would not be covered by the current setup of resampling the images to epipolar geometry (would it?). This implication should be mentioned.***"**

**Authors Response:**

We added a reference to Heinrichs and Rodehorst (2006), who gives a solution for perspective cameras. An adaption to omnidirecitonal cameras has, to the best of our knowledge, not been addressed yet, but seems to be possible, but we didn't investigated it. The revised text is in Page 13, Line 30:

"*A successful integration of a third camera into the dense stereo matching scheme including epipolar rectification is explained for perspective cameras in Heinrichs and Rodehorst (2006). An adaption to omnidirectional cameras has, to the best of our knowledge, not been addressed yet, but seems to be possible.*"

References:

[Heinrichs 2006]

Matthias Heinrichs and Volker Rodehorst. "Trinocular rectification for various camera setups." *Symp. of ISPRS Commission III-Photogrammetric Computer Vision PCV*. Vol. 6. 2006.

List of all relevant changes (Pages and Lines w.r.t. the marked-up text):

[revised manuscript text omitted]

25    ~~Next, we demonstrate the spatial variability of the 3D reconstruction error based on our real stereo setup by assuming a constant error in the disparity map. Based on the distance between our cameras, we can estimate the uncertainty of the reconstruction using the triangulation method presented in . shows the reconstruction error for a virtual cloud layer at 1500m and 3000m height assuming an error of 1 pixel in the disparity map after the matching phase, , which corresponds to a directional error of approximately 0.1°. The values represent absolute errors within the epipolar plane. Therefore, depending~~

30    ~~on the horizontal distance to the cameras, the error has a larger vertical component (small distance) or a larger horizontal component (larger distance). The error grows larger in two cases: First, with increasing distance to the cameras, and second, with increasing co-linearity of the object point and the camera centers. In both cases, the angle γ between the projection rays becomes very small, yielding larger triangulation errors. Thus, the sky imagers do not provide a hemispheric 3D reconstruction with homogeneous accuracy. However, the accuracy can be ameliorated by employing a third camera in a triangle configuration.~~

The imaging process of a sensor adds a random noise signal, which can be limited, but not avoided. In principle, this also  affects parameter estimation, because both localization and measurement are disturbed. Given a large number of measurements for the calibration, the signal noise can be compensated in a maximum likelihood estimation as the redundancy is high. The stereo matching is also  affected by the noisy image signal and causes a disturbed 3D reconstruction

5   .

 In the following analysis we investigate the effects of an asynchronous recording of the stereo images during the observation of a dynamic cloud scene.

Despite frequent requests to an NTP service, we sometimes experience asynchronous system times on the local computers in the range of a few seconds. Consequently, the whole cloud scene shifts between the single shots and thus causes a displacement

10  in the images, which leads to a biased or flawed disparity map. We investigate the effects of a cloud field displacement of $\Delta = \pm 15\,\text{m}$ along the baseline (x-direction) and perpendicular to the baseline (y-direction) in a virtual sky imager setup together with a virtual cloud layer at 3 km, using the 3D rendering software Blender (Blender Foundation, 2016). The virtual sky imagers have identical internal camera geometries comparable to the real ones, and a relative pose of $R_L = R_R = I$ and $\mathbf{t} = (300, 0, 0)^\top\,\text{m}$. Figure 13 shows the cross-sections of the respective reconstruction along the baseline (x-axis) and

15  perpendicular to the baseline (y-axis). A displacement along the x-axis results in a lower (2875 m for $\Delta = -15\,\text{m}$) or a higher (3183 m for $\Delta = +15\,\text{m}$) cloud base compared to the unaffected reconstruction (3025 m for $\Delta = 0\,\text{m}$), while a displacement along the y-axis just causes an overall higher standard deviation of the reconstruction without a systematic error in the mean base height ($\sigma(\Delta = +15\,\text{m}) = 45\,\text{m}$, $\sigma(\Delta = -15\,\text{m}) = 48\,\text{m}$ and $\sigma(\Delta = 0\,\text{m}) = 35\,\text{
[revised manuscript text omitted]